# Uncertainty of simulated groundwater recharge at different global warming levels: A global-scale multi-model ensemble study

Robert Reinecke[1,2], Hannes Müller Schmied[2,3], Tim Trautmann[2], Lauren Seaby Andersen[4], Peter Burek[5], Martina Flörke[6], Simon N. Gosling[7], Manolis Grillakis[8], Naota Hanasaki[9], Aristeidis Koutroulis[10], Yadu Pokhrel[11], Wim Thiery[12,13], Yoshihide Wada[5,14], Satoh Yusuke[5,15], Petra Döll[2,3]

[1]International Center for Water Resources and Global Change (UNESCO), Koblenz, 56002, Germany
[2]Institute of Physical Geography, Goethe University Frankfurt, Frankfurt, 60438, Germany
[3]Senckenberg Leibniz Biodiversity and Climate Research Centre (SBiK-F) Frankfurt, Frankfurt, 60325, Germany
[4]Potsdam Institute for Climate Impact Research, Telegraphenberg A31, 14473 Potsdam, Germany
[5]International Institute for Applied Systems Analysis, Schlossplatz 1, 2361 Laxenburg, Austria
[6]Facutly of Civil Engineering, Ruhr-University Bochum, Bochum, 44801 Bochum, Germany
[7]School of Geography, University of Nottingham, Nottingham NG7 2RD, United Kingdom
[8]Institute for Mediterranean Studies, Foundation for Research and Technology Hellas, Rethymno 74100, Greece
[9]National Institute for Environmental Studies, Tsukuba, 305-8506, Japan
[10]School of Environmental Engineering, Technical University of Crete, Chania 73100, Greece
[11]Department of Civil and Environmental Engineering, Michigan State University, East Lansing, Michigan 48824 USA
[12]Vrije Universiteit Brussel, Department of Hydrology and Hydraulic Engineering, Pleinlaan 2, 1050 Brussels, Belgium
[13]ETH Zurich, Institute for Atmospheric and Climate Science, Universitaetsstrasse 16, 8092 Zurich, Switzerland
[14]Department of Physical Geography, Faculty of Geosciences, Utrecht University, the Netherlands
[15]National Institute for Environmental Studies, Center for Global Environmental Research, Tsukuba, Japan

*Correspondence to*: Robert Reinecke (robert.reinecke@uni-potsdam.de)

**Abstract.** Billions of people rely on groundwater as an accessible source for drinking water and irrigation, especially in times of drought. Its importance will likely increase with a changing climate. It is still unclear, however, how climate change will impact groundwater systems globally and thus the availability of this vital resource. Groundwater recharge is an important indicator for groundwater availability, but it is a water flux that is difficult to estimate as uncertainties in the water balance accumulate, leading to possibly large errors in particular in dry regions. This study investigates uncertainties in groundwater recharge projections using a multi-model ensemble of eight global hydrological models (GHMs) that are driven by the bias-adjusted output of four global circulation models (GCMs). Pre-industrial and current groundwater recharge values are compared with recharge for different global warming (GW) levels as a result of three representative concentration pathways (RCPs). Results suggest that projected changes strongly vary among the different GHM-GCM combinations, and statistically significant changes are only computed for few regions of the world. Statistically significant GWR increases are projected for Northern Europe and some parts of the Arctic, East Africa and India. Statistically significant decreases are simulated in southern Chile, parts of Brazil, central USA, the Mediterranean, and southeast China. In some regions, reversals of groundwater recharge trends can be observed with global warming. Because most GHMs do not simulate the impact of changing atmospheric $CO_2$ and climate on vegetation and thus evapotranspiration, we investigate how estimated changes in

GWR are affected by the inclusion of these processes. In some regions, inclusion leads to differences in groundwater recharge changes of up to 100 mm year$^{-1}$. Most GHMs with active vegetation simulate less severe decreases of groundwater recharge than GHMs without active vegetation and in some regions even increases instead of decreases. However, in regions where GCMs predict decreases in precipitation and groundwater availability is most important, model agreement among GHMs with active vegetation is lowest. Overall large uncertainties in the model outcomes suggest that additional research on simulating groundwater processes in GHMs is necessary.

## 1 Introduction

The critical role of groundwater as an accessible source for irrigation and drinking water in particular during dry periods, droughts, and floods will intensify with climate change because increased precipitation variability is expected to decrease the reliability of surface water supply (Taylor et al., 2013; Döll et al., 2018; Kundzewicz and Döll, 2009). While demand for groundwater is likely to increase in the future, groundwater abstractions have already led to depleted aquifers in many regions around the globe (Thomas and Famiglietti, 2019; Cuthbert et al., 2019a; Wada et al., 2012; Konikow and Kendy, 2005; Döll et al., 2014b). They have also resulted in the reduction of groundwater discharge to rivers with negative impacts on water availability for humans and freshwater biota in particular during low-flow periods (Herbert and Döll, 2019). To what extent groundwater can serve for sustaining ecosystem health and for supporting human adaptation to climate variability and change strongly depends on future groundwater availability, which is strongly affected by climate change (Kundzewicz and Döll, 2009; Döll, 2009; Taylor et al., 2013; Cuthbert et al., 2019b).

Groundwater recharge (GWR) is a central indicator of potential groundwater availability (Herbert and Döll, 2019). GWR is the vertical water flux to the groundwater from the soil (diffuse GWR) and from surface water bodies (point or focused recharge) (Small, 2005). It is a function of the local climate, topography, soil, land cover, land use (urbanization, woodland establishment, crop rotation, and irrigation practices), atmospheric $CO_2$ concentrations, and geology (Small, 2005). Changes in GWR alter groundwater levels and their temporal patterns, which affect vital ecosystem services (Kløve et al., 2014). Knowledge of the dynamics and process interactions determining GWR is a fundamental prerequisite to assess groundwater quality and quantity under climate change (Green et al., 2011). The simulation of GWR is possibly one of the most challenging components of the water budget as it accumulates the uncertainties of all other components of the budget. Especially in semiarid regions, uncertainties in precipitation and evapotranspiration (Wartenburger et al., 2018) lead to considerable uncertainty in recharge. An additional factor in estimating groundwater recharge is the simulation of the groundwater table and thus capillary rise and focused recharge. This has not been achieved yet in GHMs, however, recently, global hydrological models (GHMs) started integrating gradient-based groundwater models to better estimate the flows between surface water and groundwater as well as the impact of humans and the changing climate on the groundwater system (de Graaf et al., 2019; Reinecke et al., 2019). Neglecting capillary rise may lead to an overestimation of decreases and increases of GWR due to a changing climate.

Assessing the response of GWR to climate change is difficult even at the local scale, one of the reasons being that groundwater recharge, different from streamflow, is rarely measured, and long time series of groundwater recharge are not available (Earman and Dettinger, 2011). In local groundwater modelling, groundwater recharge is often determined by calibration using hydraulic head observation, while integrated modelling relies on the partitioning of precipitation into evapotranspiration, storage change, and runoff (GWR plus surface and subsurface runoff). Moreover, projections of GWR

often neglect the impact of changing climate and higher $CO_2$ levels on plants and thus evapotranspiration and GWR (Taylor et al., 2013). With higher $CO_2$ levels, terrestrial plants open their stomata less, which reduces evapotranspiration and increases runoff (physiological effect) while they might grow better, increasing evapotranspiration (structural effect) (Gerten et al., 2014). Vegetation models that include these effects disagree about the balance of both effects (Gerten et al., 2014). However, based on a large ensemble of GCMs that include the impact of $CO_2$ and changing climate on vegetation and evapotranspiration,

rising $CO_2$ can be expected to decrease transpiration and thus increase total runoff (Milly and Dunne, 2016). Therefore, GHMs that do not consider active vegetation may underestimate runoff, and thus GWR increases, or they may overestimate GWR decreases.

While there have been review articles on the relation of groundwater and climate change (Smerdon, 2017; Jing et al., 2020; Refsgaard et al., 2016), global-scale studies that quantify the impact of climate change on GWR are rare. They have

evolved regarding the way climate scenarios were implemented and how many global climate models (GCMs) and GHMs were included in the study. While Döll (2009) could only use the delta change method to integrate information from two GCMs in the GHM WaterGAP (Alcamo et al., 2003; Müller Schmied et al., 2014), Portmann et al. (2013) could feed their simulations of future changes in GWR with WaterGAP directly by the bias-adjusted output with five GCMs. They found that changes in GWR increase with increasing greenhouse gas emissions. Acknowledging that not only GCMs but also GHMs contribute to

the uncertain translation of emissions scenarios to changes in GWR (Moeck et al., 2016), the study of Döll et al. (2018) included two GHMs (WaterGAP and LPJmL, Rost et al. (2008), Schaphoff et al. (2013)) driven by the bias-adjusted of four GCMs. They evaluated relative changes of GWR with climate change, which can arguably serve as a better indicator of climate change hazard than absolute changes of GWR. On the other hand, the usage of relative change led to the result that change in GWR could not be reliably computed for 55% of the global land area due to very small GWR for the reference period simulated

by LPJmL (Döll et al., 2018). While the LPJmL model considered, different from the WaterGAP model, the effect of rising $CO_2$ on groundwater recharge, the impact of this on GWR projections were not analyzed in Döll et al. (2018). In general, studies investigating the difference between GHMs with and without dynamic vegetation are rare (Davie et al., 2013).

This study assesses the impact of climate change on GWR based on the output of a multi-model ensemble encompassing eight GHMs, each forced by the bias-adjusted output of four GCMs under three different representative

concentration pathways (RCPs). The ensemble was generated in the framework of the Inter-Sectoral Impact Model Intercomparison Project (ISIMIP) using simulation protocol ISIMIP2b (Frieler et al., 2017). The ISIMIP global water sector incorporates global models, including water resources models, land surface models, and dynamic vegetation models that can compute water flows and storages on the continents of the Earth; in this study, all three model types are referred to as GHMs.

The ISIMIP2b ensemble has already been used in multiple climate change studies investigating, e.g., flood risk (Willner et al., 2018; Thober et al., 2017; Alfieri et al., 2017), low flows in Europe (Marx et al., 2018), evapotranspiration (Wartenburger et al., 2018), runoff and snow in Europe (Donnelly et al., 2017), drought severity (Pokhrel et al., 2021), heat uptake by inland waters (Vanderkelen et al., 2020) or multi-sectoral impacts (Byers et al., 2018; Lange et al., 2020).

We analyze how GWR is projected to change globally and regionally for multiple global warming (GW) levels, determine the contributions from GHMs and GCMs to the variance of simulated changes and discuss the implications for future assessments of global groundwater resources. Furthermore, we show the effect of including the physiological impacts of evolving $CO_2$ on global estimates of GWR. To this end, the remainder of this paper is structured as follows. Section 2 provides an overview of the used GHMs and the methods to calculate changes of GWR per GW level and sources of uncertainty. The results in section 3 show the significant changes in GWR per GW and the differences in between GHMs and GCMs. We then compare the influence of GCMs, GHMs, and RCPs on the variance of simulated GWR, assess the differences in GWR due to including dynamic vegetation in GHMs and compare the GHM simulations to interpolated measured GWR. The paper closes with a discussion of these findings (Sect. 4) and conclusions (Sect. 5).

## 2 Methods

### 2.1 Simulation of groundwater recharge

This study encompasses eight GHMs that differ in their representation of various hydrological processes. Four of these models are able to simulate the impact of evolving $CO_2$ concentrations on vegetation: CLM 4.5, JULES-W1, LPJmL, MATSIRO (Table 1). In the following, we use the term *active vegetation* for models that consider the physiological effect of changes in $CO_2$ on vegetation and the term *dynamic vegetation* for the models that allow for changing vegetation regarding LAI and/or vegetation type. A comprehensive overview of GHMs and their properties can be found in Sood and Smakhtin (2014). Detailed model descriptions and evaluations of the models can be found in the primary publications referred to in the subsections below and Telteu et al. (2021) (for the model parameterisation see Sect. 2.2.). The definition of GWR and groundwater varies in between GHMs (discussed in Sect. 4). The analysis in this study is based on monthly GWR (variable *qr* in ISIMIP) in 0.5° x 0.5° grid cells simulated by the eight GHMs taking part in the ISIMIP2b protocol (Frieler et al., 2017). Some GHMs contained small negative GWR values, e.g., due to capillary rise; these values were set to zero in the analysis. We do not consider focused recharge in this study as no model offers a reliable implementation of these processes until now. Also, none of the models simulate he depth of the groundwater table beneath the land surface which does not allow to correctly attribute delays in recharge due to water table depth.

**Table 1** Overview which models are able to simulate the impact of evolving $CO_2$ concentrations on vegetation and how it is implemented.

| GHM | Considers $CO_2$ | Summary of considered vegetation processes in ISIMIP2b | Reference |
|---|---|---|---|
| WaterGAP2 | No | - | - |
| CLM4.5 | Yes | Photosynthesis depends on root zone soil moisture availability. The description is similar to LPJmL listed below. The area a population of plant functional types (PFTs) takes up is prescribed and only changes if the input data changes. | (Di Liu and Mishra, 2017) |
| H08 | No | - | - |
| JULES-W1 | Yes | Evapotranspiration is considered from five PFTs and four non-vegetative surface types. Each grid cell is composed of different fractions of those nine surface types. Transpiration occurring from vegetation is based on photosynthetic process, which is subject to stomatal conductance regulated by the $CO_2$ concentration. Furthermore, transpiration is also controlled by soil moisture availability in the root zone. | (Best et al., 2011; Clark et al., 2011) |
| LPJmL | Yes | Vegetation composition is determined by the fractional coverage of PFTs at the grid-scale. PFTs are defined to account for the variety of structure and function within a stand and are therefore simulated as average individuals competing for light and water according to their crown area, LAI, and rooting profiles. The vegetation dynamics component of LPJmL includes carbon allocation to different PFT tissue compartments, PFT interaction, and establishment and mortality processes. Photosynthesis and stomatal response are simulated following Farquhar et al. (1980) and the generalization by Collatz et al. (1991) for global modelling, based on the function of absorbed photosynthetically active radiation, temperature, day-length, and canopy conductance for each PFT present in a grid cell. | (Schaphoff et al., 2018) |
| PCR-GLOBWB | No | - | - |
| CWatM | No | - | - |
| MATSIRO | Yes | The consideration of $CO_2$ effects is functionally similar to that in CLM, and there is no dynamic vegetation scheme. $CO_2$ is prescribed in the model, which is used in the photosynthesis scheme to calculate stomatal conductance, among other parameters, following Farquhar et al. (1980). Soil moisture stress on photosynthesis is considered using moisture availability in the root zone with root distribution fraction in each soil layer. All of that is done for different vegetation or plant functional types. | (Takata et al., 2003) |

## WaterGAP2

The WaterGAP2 model (Alcamo et al., 2003) computes human water use in five sectors and the resulting net abstractions from groundwater and surface water for all land areas of the globe, excluding Antarctica. These net abstractions are then taken from the respective water storages in the WaterGAP Global Hydrology Model (WGHM) (Müller Schmied et al., 2014; Döll et al., 2003; Döll et al., 2012; Döll et al., 2014b). With daily time steps, WGHM simulates flows among the water storage compartments canopy, snow, soil, groundwater, lakes, human-made reservoirs, wetlands, and rivers. GWR in WaterGAP2 is calculated as a fraction from runoff from land-based on soil texture, relief, aquifer type, and the existence of permafrost or glaciers, taking into account a soil texture dependent maximum daily groundwater recharge rate (Döll and Fiedler, 2008). If a grid cell is defined as semiarid/arid and has a medium or coarse soil texture, GWR will only occur if daily precipitation exceeds a critical value (Döll and Fiedler, 2008); otherwise, the water runs off. Runoff from land that does not contribute to GWR is transferred to surface water bodies as fast surface runoff. WaterGAP further computes focused recharge beneath surface water bodies in semiarid/arid grid cells, which is not considered in this study.

## CLM4.5

The Community Land Model version 4.5 (CLM4.5) (Lawrence et al., 2011; Oleson et al., 2013; Swenson and Lawrence, 2015) is the land component of the Community Earth System Model (CESM), a fully-coupled, state-of-the-art earth system model (Hurrell et al., 2013). CLM is a land surface model representing the physical, chemical, and biological processes through which terrestrial ecosystems influence and are influenced by climate, including $CO_2$, across a variety of spatial and temporal scales (Lawrence et al. 2011). Individual land grid points can be composed of multiple land units due to the nested tile approach, which enables the implementation of multiple soil columns and represents biomes as a combination of different plant functional types. Groundwater processes, including sub-surface runoff, recharge, and water table depth variations, are simulated based on the SIMTOP scheme (Niu et al., 2007; Oleson et al., 2013).

## H08

H08 (Hanasaki et al., 2018) is a GHM including various components for water use and management. It consists of five major components, namely, a simple bucket-type land surface model, a river routing model, a crop growth model which is mainly used to estimate the timing of planting, harvesting, and irrigation in cropland, a reservoir operation model, and a water abstraction model. The abstraction model supplies water to meet the daily water demand of three sectors (irrigation, industry, municipality) from six available and accessible sources (river, local-reservoir, aqueduct, seawater desalination, renewable groundwater, and non-renewable groundwater) and one hypothetical one termed unspecified surface water. It has two soil layers; one is to represent the unsaturated root zone, and the other the saturated zone (groundwater). The scheme of GWR computation is identical to Döll and Fiedler (2008).

## JULES-W1

The Joint UK Land Environment Simulator (JULES) (Best et al., 2011) (W1 stands for water-related simulations in the ISMIP framework) is a land surface model initially developed by Met Office as the land surface component of Met Office Unified Model. JULES is a process-based model that simulates the carbon, water, energy, and momentum fluxes between land and atmosphere, including plant - carbon interactions (Clark et al., 2011). The rainfall that reaches the ground is partitioned into hortonian surface runoff and an infiltration component. Four soil layers represent the soil column with a total thickness of 3 m, with a unit hydraulic head gradient lower boundary condition, and no groundwater component. The water that infiltrates the soil moves down the soil layers updated using a finite difference form of the Richards equation (Best et al., 2011). The saturation excess water from the bottom soil layer becomes subsurface runoff that can be considered to be GWR (Le Vine et al., 2016).

## LPJmL

Lund Potsdam Jena managed Land (LPJmL) is a dynamic global vegetation model that simulates the growth and productivity of both natural and agricultural vegetation as coherently linked through their water, carbon, and energy fluxes  (Schaphoff et al., 2018). The soil column is divided into six active hydrological layers with a total thickness of 13 m depth. Percolation of infiltrated water through the soil column is calculated according to a storage routine technique that simulates free water in the soil bucket (Arnold et al., 1990). Excess water over the saturation levels produces lateral runoff in each layer (subsurface runoff). GWR is considered to be percolation (seepage) from the bottom soil layer. As there is no groundwater storage in LPJmL, for the ISIMIP2b protocol, seepage from the base soil layer is reported as both GWR and groundwater runoff, which is routed directly (no time delay) back into the river system.

## PCR-GLOBWB

PCR-GLOBWB (PCRaster Global Water Balance; (Sutanudjaja et al., 2018) simulates the water storage in two vertically stacked soil layers and an underlying groundwater layer. Water exchanges are simulated in-between the layers (infiltration, percolation, and capillary rise) as well as the interaction of the top layer with the atmosphere (rainfall, evapotranspiration, and snowmelt). PCR-GLOBWB also calculates canopy interception and snow storage. Natural groundwater recharge is fed by net precipitation, and additional recharge from irrigation occurs as the net flux from the lowest soil layer to the groundwater layer, i.e., deep percolation minus capillary rise. The ARNO scheme (Todini, 1996) is used to separate direct runoff, interflow, and GWR. Groundwater recharge can be balanced by capillary rise if the top of the groundwater level is within 5 m of the topographical surface (calculated as the height of the groundwater storage over the storage coefficient on top of the streambed elevation and the sub-grid distribution of elevation).

**CWatM**

The Community Water Model (CWatM) is a large-scale integrated hydrological model, which encompasses general surface and groundwater hydrological processes, including human hydrological activities such as water use and reservoir regulation (Burek et al., 2019). CWatM takes six land cover classes into account and applies the tile approach. This hydrological model has three soil layers and one groundwater storage. Depth of the first soil layer is 5 cm, and the depth of second and third layers vary over grids depending on the root zone depth of each land cover class, resulting in total soil depth of up to 1.5 m.

Groundwater storage is designed as a linear reservoir. CWatM includes preferential bypass flow directly into groundwater storage and capillary rise from groundwater storage, as well as percolation from the third soil layer to groundwater storage. Hence, the groundwater recharge reported by CWatM in ISIMIP2b is the net recharge calculated from these three terms.

**MATSIRO**

The Minimal Advanced Treatments of Surface Interaction and RunOff (MATSIRO; Takata et al. (2003)) is a global land

surface model initially developed for an Atmospheric Ocean General Circulation Model, the Model for Interdisciplinary Research On Climate (Hasumi, H., and S. Emori, 2004). This process-based model calculates water and energy flux and storage at and below the land surface, considering the stomatal response to $CO_2$ increase as well in the photosynthesis process. The off-line version of MATSIRO used for ISIMIP2b simulation explicitly takes vertical groundwater dynamics into account, including groundwater pumping (Pokhrel et al., 2015; Pokhrel et al., 2012). Soil moisture flux between the 15 soil layers is

expressed as a function of the vertical gradient of the hydraulic potential, which is the sum of the matric potential and the gravitational head, and soil moisture movement is calculated by Richards equation. MATSIRO calculates net groundwater recharge as a budget of gravitational drainage into and capillary rise from the layer where the groundwater table exists. A simplified TOPMODEL (Beven and Kirkby, 1979; Stieglitz et al., 1997) is used to represent surface runoff processes, and groundwater discharge is simulated by using an unconfined aquifer model (Koirala et al., 2014).

**2.2 Model simulations**

Each GHM is forced by bias-adjusted data from four GCMs: GFDL-ESM2M, HadGEM2-ES, IPSL-CM5A-LR, and MIROC5. Further details on the selection of climate models and the bias correction can be found in Frieler et al. (2017), Lange (2016), Hempel et al. (2013), Lange (2018), and online at ISIMIP (2018). The bias adjustment method used for the GCMs in ISIMIP2b is using a trend preserving algorithm (Frieler et al., 2017) with EWEMBI (Lange 2018) as baseline (reference) climate

condition. The simulations in this study span the period 1861 till 2099. All GHMs (except for PCR-GLOBWB, which misses the RCP 8.5 run) simulate the RCPs 2.6, 6.0, and 8.5.

The pre-industrial period (PI) is defined in ISIMIP from 1661-1860, whereas the historical period is defined from 1861-2005. Additionally, to the RCP and historical simulations, ISIMIP defines PI simulations that represent an extended state of emissions scenarios from the PI period till 2099 (and partially till 2300, not applicable in this study). In this study, we

always, if not stated otherwise, refer with PI to the simulation period 1960-2099 with the continued concentration levels of 1661-1860. Details on the simulation setup can be found on the ISIMIP webpage ISIMIP (2019) or in Frieler et al. (2017).

Regarding the non-climatic drivers, all GHMs use, for the time before 2006, so-called historical socio-economic pathway assumptions, e.g., historical water use, except for CLM 4.5, which used the socio-economic state of 2005. All simulations for 2006-2099 are based on this assumed socio-economic state of 2005. For some models this affects the
abstraction from groundwater, which is not stimulated by all models (JULES-W1), or GWR directly due to irrigation (H08, CLM, PCR-GLOBWB). Details on the pertinent scenario variables can be found in the ISMIP protocol (Frieler et al., 2017). Land-use change was not considered.

## 2.3 Determining stabilized warming levels

In order to derive policy-relevant information, we assed impacts framed in terms of GW levels (1°, 1.5°, 2°, and 3°C) with
respect to the GW of 0°C in PI conditions (James et al., 2017). The time of passing a warming level is defined as the first time the 31-year running mean of the global averaged annual mean temperature gets above that level. Each GCM reaches different GW at different times (Table 2), depending on the RCPs (van Vuuren et al., 2014). For each GW level (1°, 1.5°, 2°, and 3°C), time slice of 31 years (15 before the level was reached, and 15 after) for each GCM and for each RCP, in which that GW is reached, are used. Using this time slice, a yearly mean GWR at 0.5° was calculated for the GHMs that were forced with the
particular combination of GCM and RCP (Fig. 1). Additionally, a PI reference was calculated for each GCM, RCP, and GHM combination for the same time-slice the GW level was reached in a particular RCP-GCM combination using the PI reference simulation (see section 2.2). Figure 1 illustrates the methodology by showing two unspecified RCPs and the PI comparison paths.

Considering that not all RCP/GCM combinations reach higher warming levels (Table 2), not all ensembles have the
same size. Theoretically, the maximum ensemble size is 96, a combination of 8 GHMs, 4 GCMs, and 3 RCPs (2.6, 6.0, and 8.5). Because projections under RCP 8.5 were not available for PCR-GLOBWB, the maximum ensemble size is 84. The smallest ensemble (for 3°C) consists of 36 members.

**Table 2** Overview of the warming levels and in which year they are reached in the corresponding GCM (ISIMIP, 2019).

| Warming Level | RCP | GFDL-ESM2M | HadGEM2-ES | IPSL-CM5A-LR | MIROC5 |
|---|---|---|---|---|---|
| 1° | 2.6 | 2014 | 2012 | 1993 | 2015 |
| | 6.0 | 2016 | 2014 | 1993 | 2023 |
| | 8.5 | 2014 | 2012 | 1993 | 2014 |
| 1.5° | 2.6 | - | 2026 | 2009 | 2048 |
| | 6.0 | 2056 | 2032 | 2010 | 2052 |
| | 8.5 | 2036 | 2025 | 2009 | 2033 |

| | | | | | |
|---|---|---|---|---|---|
| 2° | 2.6 | - | - | 2029 | - |
| | 6.0 | 2076 | 2050 | 2029 | 2071 |
| | 8.5 | 2053 | 2037 | 2024 | 2048 |
| 3° | 2.6 | - | - | - | - |
| | 6.0 | - | 2076 | 2068 | - |
| | 8.5 | 2082 | 2056 | 2046 | 2071 |


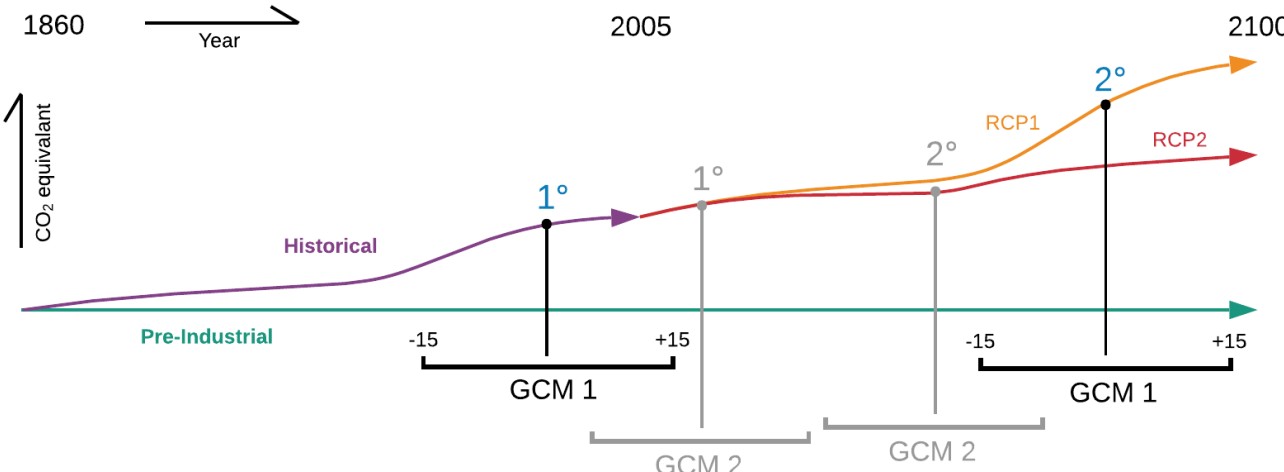

**Figure 1** Conceptual representation of how GW levels are determined for different GCMs, RCPs, and the PI comparison period.

**2.4 Calculation of model variance**

To calculate whether the variance in absolute GWR change is mainly introduced through the GHMs or the GCMs, the
following equation was applied per model grid cell and GW level.

$$Rvar_{GWR}^{model} = \sigma_{GWR}^2(GCMs) \Big/ (\sigma_{GWR}^2(GCMs) + \sigma_{GWR}^2(GHMs)) \qquad (1)$$

where $Rvar_{GWR}^{model}$ is the variance ratio of GCMs to GHMs, $\sigma_{GWR}^2(GHMs)$ is the average variance of GWR change of all GHMs
per GCM per RCP, and $\sigma_{GWR}^2(GCMs)$ is the average variance in GWR change of all GCMs per RCP per GHM. The variance
relative to the choice in RCP $Rvar_{GWR}^{RCP}$ can be calculated similarly as

$$Rvar_{GWR}^{RCP} = \sigma_{GWR}^2(RCPs) \Big/ (\sigma_{GWR}^2(RCPs) + \sigma_{GWR}^2(GHMs)) , \qquad (2)$$

where $\sigma_{GWR}^2(RCPs)$ is the average variance in GWR of all RCPs per GCM per GHM.

## 2.5 Determining significant changes

A model ensemble allows us to consider the uncertainty in modeling physical processes as different model use different algorithms and parameters for computing groundwater recharge. To determine whether changes in GWR due to GW computed by the model ensemble are statistically significant, we used the two-sample Kolmogorov–Smirnov (K-S) test to compare the GWR values computed by all GHM-GCM model combinations under e.g., PI conditions with the values at the various GW levels. The use of a two-tailed t-test is not advisable in this setting due to the small sample size (max. 84 in this study). Because the K-S test does not allow to check whether the ensemble agrees on the sign of change in GWR, we applied an additional criterion to determine a significant change similar to Döll et al. (2018). A change is only marked as statistically significant if the K-S test indicates a significant difference and at least 60% of the model realizations of the ensemble (RCP, GCM and GHM combinations) agree on the sign of change (i.e. a decrease or increase). In case of a low significance, all models may show large responses to climate change while their agreement on the amount or sign of change is low.

## 3 Results

### 3.1 Changes of groundwater recharge at different warming levels

To assess the impact of GW on GWR, Fig. 2 shows the ensemble mean change of GWR between the current 1°C world and a potential 3°C GW. We chose to express changes as absolute change rather than relative change because zero, or close to zero, GWR in some regions of the world leads to not defined or extremely large percentage increases and decreases (Fig. S1 and S2). The model mean shows large decreases of over 100 mm year$^{-1}$ in South America and in the Mississippi Basin and decreases

of up to 50 mm year⁻¹ in the Mediterranean, East China, and West Africa. Increases of over 100 mm year⁻¹ are prominent in

Indonesia and East Afrika. Individual GHM-GCM model combinations compute much larger changes.

3° compared to present day (1°) [ensemble mean]

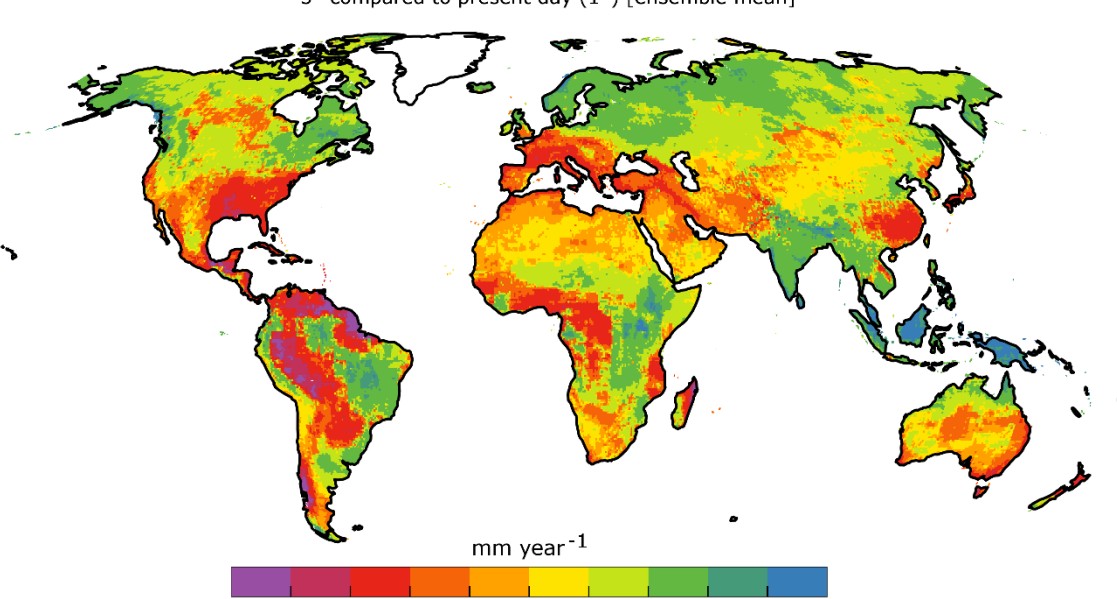

**Figure 2** Ensemble mean change in GWR [mm year-1] between conditions of present day warming of 1 °C GW and at 3 °C GW, averaged over the GWR changes of all GHM-GCM model combinations.

Ensemble mean changes as shown in Fig. 2 may be low in some areas, but this could be due to large positive changes compute

by some GHM-GCM model combinations being cancelled by large negative changes by other model combinations. To assess the changes which show a high statistical agreement in-between the model combinations, we determine where computed changes of GWR are statistically significant (Section 2.5). As a reference for the intensity of the changes, Figure 3a shows the mean GWR at PI averaged over all GHMs, RCPs, and GCMs from 1861-2099. The spatial pattern of GWR roughly agrees with the pattern of Mohan et al. (2018) derived by inferring it from more than 700 small-scale GWR estimates. The global

mean GWR for the PI period is 140 mm year⁻¹, which is very similar to the value of 134 mm yr⁻¹ determined by Mohan et al. (2018) for the period 1981-2014 (see also Sect. 4).

Figure 3b-e show the (statistical) significant (bright colors, Sect. 2.5) mean absolute changes in GWR of the multi-model ensemble under a GW of 1.0°C, 1.5°C, 2.0°C, and 3.0°C compared to PI, i.e., GWR of the PI runs for the corresponding time-slices (Sect. 2.3). For all GW levels compared to PI (Figure 3b-e), consistent patterns of decreasing GWR emerge for

southern Chile, Brazil, central continental USA, the Mediterranean, and East China. Consistent and significant increases can be observed for northern Europe and in general northern latitudes and East Africa. Significant changes could only be derived for a small percentage of the total grid cells. Only about 15% of the cells, on average for all GW levels, show significant increases or decreases. However, the patterns of non-significant (light colors) mean changes are consistent with the significant

changes and show, e.g., for the Amazon larger areas of increases and decreases around the significant changes. The identification of non-significance in most areas is due to the K-S test. The sign criterion affects mainly the Sahara and Central Asia.

At 1°C GW (Figure 3b), decreases of more than 100 mm year$^{-1}$ are simulated in Southeast Asia, East China, Guyana, and southern Brazil. Decreases between 100 and 50 mm year$^{-1}$ can be seen in central continental USA, southern Brazil, southern Chile, the Mediterranean, central Africa, and East China. Increases in GWR of 50 and over 100 mm year$^{-1}$ are visible in the center of the Amazon while decreases show in the northeast and southern part that increase with GW. Overall, the significant global change is -17 mm year$^{-1}$ at 1°C.

A 1.5°C GW shows only a limited increase in the Amazon but similar increases in the rest of the world. Decreases in GWR over 100 mm year$^{-1}$ are now visible in Central America, but decreases for Southeast Asia have vanished. Smaller decreases, for example, in Australia, have vanished as well in a 1.5°C world. These effects are not necessarily due to no changes in GWR but due to disagreements in the ensemble that do not allow to determine a reliable and significant change for this warming level. The global significant mean change is -12 mm year$^{-1}$ at 1.5°C GW.

At 2°C GW, increases in GWR over 100 mm year$^{-1}$ are present in northern Java, Amazon, and East Africa. Decreases are similar to 1.5°C GW, except for southern Chile and the northern Andes, where decreases become more severe. However, on the significant global mean, these changes balance out to -1 mm year$^{-1}$.

In a 3°C world, large areas of decreases in GWR of over 100 mm year$^{-1}$ in the Amazon Basin close to the Andes occur, also in Guyana, Venezuela, West Africa, and the Mississippi Basin. Increases in GWR of over 100 mm year$^{-1}$, in contrast, are visible in East Africa, India, and North Java. Increases of 50 to 100 mm year$^{-1}$ dominate in northern latitudes at 3 °C warming compared to other GW levels. The global significant mean increases by +3 mm year$^{-1}$.

We have already reached a GW of approximately 1°C (IPCC, 2018). Figure 3f shows the changes in GWR of a 3° GW compared to the present-day GW of already 1°C instead of the PI. Overall, the agreement among the models is smaller than when the 3°C world is compared to PI. Only 8% of the cells show significant changes. Decreases over 100 mm year$^{-1}$ are present in the Amazon Basin close to the Andes and on the coast of Guyana. Decreases of 50 to 100 mm year$^{-1}$ are visible in Chile, the Mississippi Basin, the Caribbean, and southern France. Increases in GWR are again to be expected in the northern Latitudes, southern Brazil, East Africa, and Southeast Asia, whereas the latter shows increases over 100 mm year$^{-1}$ for Malaysia. The global significant mean change is +8 mm year$^{-1}$. Figure S3 shows the mean and median changes of GWR per latitude for all four GW levels, together with the standard deviation without a significance test. A decrease in mean GWR can be observed for all GW levels at 40° S, around 20° S (Namibia, Australia), and 5° N (Guyana). Increases are visible at 60° N (North Europe) and southerly close to the Equator, presenting a large spread and sudden change in directions in the tropics. Increases at greater than 60° N are likely due to a combination of different rain and snow patterns as well as snowmelt timing.

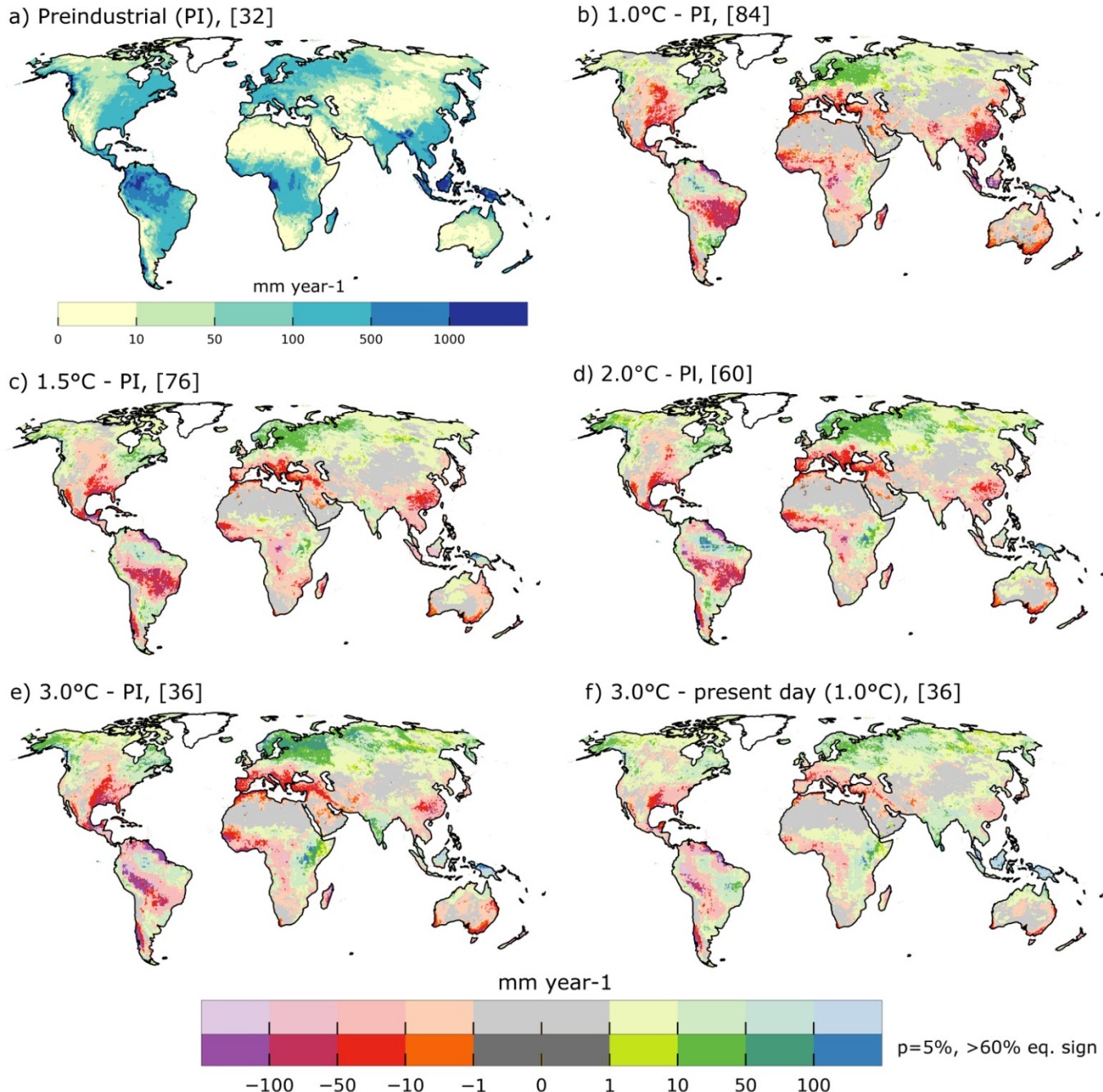

**Figure 3** Mean GWR [mm year⁻¹] for pre-industrial greenhouse gas concentrations, averaged over the GWR of all GHMs and GCMs (a). Ensemble mean absolute change in GWR [mm year⁻¹] at 1.0 °C (b), 1.5°C (c), 2.0°C (d), and 3.0°C (e) GW compared to PI. The ensemble mean absolute change in GWR [mm year⁻¹] for 3.0°C GW compared to GWR at the current GW of 1°C (f). For (b) to (f) only those cells are displayed in solid colors where the Kolmogorov-Smirnov (K-S) test with a p of 5% indicated that the ensemble GWR distribution for PI (for (f) the GWR distribution at 1°C) and for the GW level differ, and at least 60% of the models agree on the sign of the change. The ensemble size is shown in brackets. Lighter colors (upper color bar) show (statistical) insignificant mean differences.

Large areas of insignificant changes of GWR (light colors) in Fig. 3 can be traced back to the uncertainty in GWR in between GHMs and GCMs. Figure 4 shows absolute GWR changes in a 1.5 °C world compared to PI (Fig. 3a,b) as well as PI GWR (Fig. 3c,d) for the SREX (Special Report on Managing the Risks of Extreme Events and Disasters to Advance Climate Change Adaptation, Murray and Ebi (2012)) region Amazon (left) and South Europe/Mediterranean (right). Corresponding plots for all other SREX regions are provided in the supplement. Similar to box plots, the letter-value plots in Fig. 4 show the distribution of values among the 0.5° grid cells belonging to the SREX region. Letter-value plots have the advantage of showing the distribution of values outside of the usual interquartile range (IQR, Q25 - Q75). For example, for Fig. 4b CLM 4.5 with GFDL-ESM2-ES, the mean change in GWR is -19 mm year$^{-1}$, the middlebox represents the IQR showing that 50% of changes are close to zero or smaller than zero, the smaller box towards the negative changes shows that 12.5% are smaller than -47 mm year$^{-1}$, whereas the additional missing box in the positive direction hints that almost no values are larger than zero. The horizontal size of the boxes is automatically scaled and does not carry any additional information.

Computed changes vary strongly among both GHMs and GCMs (Fig. 4a,b). In the Amazon, Jules-W1 shows a mean increase of 225 mm year$^{-1}$. Compared to WaterGAP2, Jules-W1 estimates of GWR change are 147 mm year$^{-1}$ higher for MIROC5 and 44 mm year$^{-1}$ lower for HadGEM. These differences are even large relative to the higher mean PI GWR in the Amazon compared to other regions of the world (compare to MED in Fig. 4). Nevertheless, also the PI estimates differ by, e.g., 122 mm year$^{-1}$ between Jules-W1 and WaterGAP2 on the mean for all GCMs and RCPs, and PI GWR is 625 mm year$^{-1}$ smaller for H08 than for MATSIRO in the Amazon.

In the Mediterranean, almost all GHMs show the largest decreases in GWR with IPSL-CM5a-LR, followed by GFDL input, while HadGEM results in almost no change. However, the changes computed with each GCM input vary strongly among the GHMs. In general, CLM 4.5 and PCR-GLOBWB project the most considerable changes. The decrease of GWR computed by CLM 4.5 with IPSL-CM5a-LR is 33% of the mean GWR calculated for PI with that model combination.

Conversely, JULES-W1 simulates for most grid cells in this SREX region the smallest PI GWR values (but also very high outliers), and likely related, the smallest (mean) changes, together with MATSIRO and CWatM, which show altogether small GWR changes in all grid cells of the SREX regions. H08 and WaterGAP2, which apply similar approaches to modeling GWR as a function of total runoff, show somewhat similar GWR changes.

The four GHMs that take into account the impact of increasing $CO_2$ (Sect. 2.1) do not result in similar changes as compared to the other four models. It is to be expected from literature (Davie et al., 2013) that with the physiological effect, the decreases of GWR would be slighter in the case of the $CO_2$-sensitive models, but that is not the case. This is likely due to the approach of analyzing GW levels instead of RCPs and periods because different GCMs reach a particular GW level at different times and $CO_2$ levels. This is further investigated in Sect. 3.3. On the global mean and for 1.5°C GW LPJmL simulates the lowest PI GWR, whereas MATSIRO and CLM 4.5 produce the highest global mean GWR (Fig. S4). PCR-GLOBWB simulates the largest global mean decreases with HadGEM (Fig. S5). In contrast, JULES-W1 and MATSIRO simulate increases of GWR on the global mean for all GCMs except for HadGEM (Fig. S5).

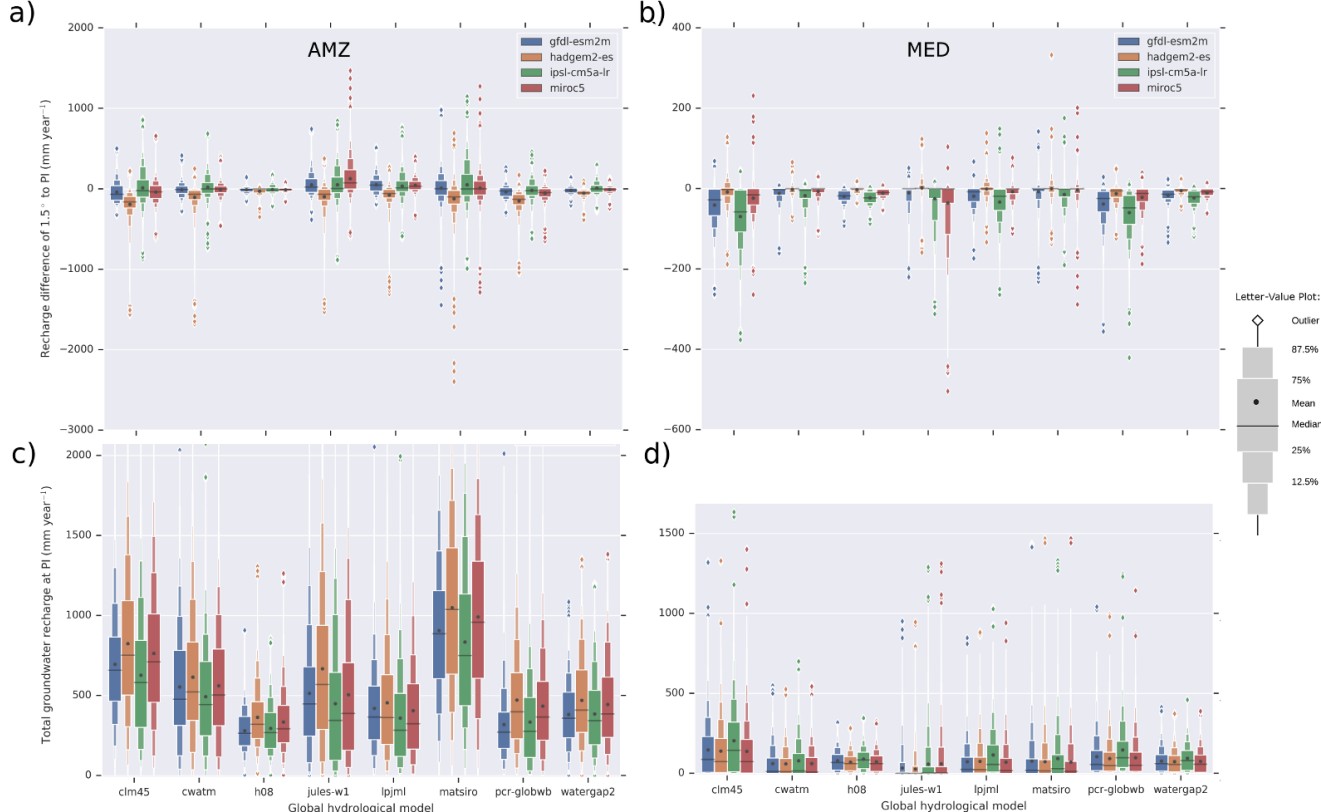

**Figure 4** Letter-value plot (Hofmann et al., 2017) of absolute changes in GWR in 0.5° grid cells [mm year⁻¹] at 1.5°C GW compared to PI (a, b) and absolute PI GWR [mm year⁻¹] (c, d) for the Amazon (a, c) and South Europe/Mediterranean (b, d) SREX region (for all other regions and GW levels [2°C, 3°C] see supplement). No statistical test is applied and all grid cells inside a region are included. Each box may include multiple simulations with different RCPs.

To provide an overview of changes in GWR in each SREX region, Table 3 shows the median, mean and $P_{25}$ and $P_{75}$ changes in GWR compared to PI for all regions (see Fig. S6 for a map of the SREX regions). Overall, North Europe shows the largest consistent increases in GWR, whereas the Amazon shows the largest consistent decreases, except for 2°C, where South Europe/Mediterranean shows the largest decreases of 18.6 mm year⁻¹ as the median. For 3°C, the Amazon shows the highest decreases in GWR of -41.0 mm year⁻¹ as median. Notably, Southeast Asia is first showing decreases of 13.1 mm year⁻¹ with 1.0°C GW and then no change with 1.5°C and 2°C and an increase in GWR of 13.5 mm year⁻¹ with 3°. Relative to PI the changes of the 3°C GW in the Amazon only account for 10% of the GWR, compared to the 19% relative increase of GWR in North Europe with 3°C and the 40% decrease in GWR in South Europe/Mediterranean at 2°C GW.

**Table 3** Median ($\widetilde{X}$), mean ($\overline{X}$), $P_{25}$, and $P_{75}$ of absolute GWR change [mm year⁻¹] for four warming levels for each SREX region compared to PI. $\widetilde{X}$, $\overline{X}$, $P_{25}$, and $P_{75}$ describe the distribution of changes of spatially averaged GWR in each SREX region among all 36-84 ensemble members (Sect. 2.3). $P_{25/75}$ are the 25th and 75th percentile in the ensemble for a given region and a given GW level. The last column shows absolute GWR at PI. The following regions are not included due to the coarse spatial resolution of the models and low confidence in the

 reliability of results: Artic, Canada/Greenland/Island, Antarctic, Pacific islands, Southern tropical pacific, Small Island Region Caribbean, West Indian Ocean. In bold maximum and minimum values per GW level. No statistical test is applied to filter the values.

| SREX | Name | 1.0° $\tilde{X}, \bar{X}$ $P_{25}, P_{75}$ | 1.5° $\tilde{X}, \bar{X}$ $P_{25}, P_{75}$ | 2.0° $\tilde{X}, \bar{X}$ $P_{25}, P_{75}$ | 3.0° $\tilde{X}, \bar{X}$ $P_{25}, P_{75}$ | PI $\tilde{X}, \bar{X}$ $P_{25}, P_{75}$ |
|---|---|---|---|---|---|---|
| AMZ | Amazon | -10.7, -14.5 -30.4, -6.8 | **-19.1**, -22.3 -38.3, -9.7 | -14.6, -18.2 -34.5, 3.4 | **-41.0**, -59.9 -81.1, -39.2 | 409.6, 550.4 419.7, 614.6 |
| CAM | Central America/Mexico | -2.4, -17.1 -23.1, -6.5 | -4.8, -21.0 -26.8, -9.0 | -4.3, -12.9 -18.9, -7.7 | -10.0, -36.0 -45.8, -24.0 | 79.8, 280.4 222.3, 327.7 |
| CAS | Central Asia | 0.0, -0.4 -0.7, 0.3 | 0.0 0.0 -0.7, 1.0 | 0.0, -0.8 -1.4, -0.3 | 0.0, -2.6 -3.9, -1.4 | 1.8, 25.9 17.2, 37.2 |
| CEU | Central Europe | 4.1, 6.8 0.5, 13.3 | 1.2, 3.1 -5.5, 11.8 | -0.4, 0.1 -9.7, 11.3 | 0.1, 2.8 -9.9, 22.3 | 114.6, 135.4 117.9, 155.8 |
| CAN | Central North America | -6.5, -16.7 -20.2, -12.3 | -5.6, -18.3 -20.2, -12.7 | -3.3, -16.6 -20.0, -12.5 | -9.9, -30.5 -32.8, -18.2 | 98.1, 128.6 76.4, 183.5 |
| EAF | East Africa | 0.0, -0.8 -2.7, 3.3 | 0.0, 2.7 -0.2, -7.8 | 0.0, 8.1 1.2, 13.9 | 0.6, 23.3 9.0, 32.4 | 32.2, 95.0 63.4, 134.1 |
| EAS | East Asia | -0.5, -15.7 -20.0, -8.3 | 0.0, -13.9 -16.9, -6.8 | 0.0, -10.3 -10.7, -3.7 | 0.0, -13.7 -14.2, -4.5 | 50.5, 147.3 113.1, 154.3 |
| ENA | East North America | 3.3, 4.8 -2.0, 11.2 | 9.9, 11.9 -0.8, 19.8 | 10.6, 15.9 -1.5, 26.3 | 1.4, 2.5 -9.1, 20.5 | 221.8, 257.8 167.4, 338.1 |
| NAS | North Asia | 0.4, 6.0 3.0, 7.2 | 0.5, 7.9 5.1, 9.1 | 3.1, 12.5 9.0, 13.1 | 4.6, 18.5 13.0, 20.4 | 24.2, 59.2 46.2, 73.4 |
| NAU | North Australia | 0.0, -4.5 -6.9, -2.2 | 0.0, -2.7 -3.9, -0.8 | 0.0, 1.1 -0.8, 3.5 | -0.9, -3.0 -7.1, 0.0 | 5.9, 43.1 28.5, 52.1 |
| NEU | North Europe | **13.1**, 24.9 15.9, 35.7 | **13.9**, 27.7 14.7, 41.3 | **18.6**, 34.9 16.8, 53.0 | **29.2**, 51.6 25.0, 78.2 | 154.8, 226.4 182.1, 280.4 |
| NEB | North-East Brazil | -8.9, -30.3 -35.6, -21.2 | -10.5, -22.9 -31.3, -13.2 | -6.2, -14.4 -24.9, -2.1 | -6.0, -9.4 -20.7, 2.1 | 161.6, 227.4 147.1, 315.0 |
| SAH | Sahara | 0.0, -0.7 -1.0, -0.3 | 0.0, 0.3 0.1, 0.4 | 0.0, -0.2 -0.2, 0.0 | 0.0, -0.4 -0.5, 0.0 | 0.1, 4.2 0.8, 4.4 |
| SAS | South Asia | -3.3, -13.4 -15.9, -8.3 | 0.0, -4.8 -6.1, 0.1 | -2.3, -11.6 -17.5, -5.3 | 3.8, 26.9 2.3, 45.5 | 151.8, 274.9 229.5, 319.2 |
| SAU | South Australia/New Zealand | -2.9, -8.6 -11.1, -4.5 | -2.3, -10.3 -12.4, -6.5 | -2.1, -15.3 -17.8, -9.4 | -4.2, -20.0 -22.2, -14.3 | 18.1, 135.7 111.4, 157.6 |
| MED | South Europe/Mediterranean | -3.9 -14.3 -17.6, -9.3 | -6.3, -18.1 -21.6, -12.8 | **-16.8**, -23.7 -27.4, -16.8 | -12.5, -28.9 -31.8, -19.1 | 43.9 84.9 72.1, 87.6 |
| SEA | Southeast Asia | **-13.1**, -36.1 -55.7, -10.7 | -0.1, -5.2 -18.0, 8.6 | -0.6, 23.1 -1.7, 36.5 | 13.5, 46.1 3.0, 68.9 | 547.9, 725.2 528.0, 881.2 |
| SSA | Southeastern South America | 0.0, -6.3 -8.3, -5.1 | 0.0, -5.2 -8.9, -4.4 | 0.0, -9.4 -12.9, -4.5 | -1.4, -11.8 -15.7, 0.3 | 61.0, 129.5 87.9, 164.6 |

| | | | | | | |
|---|---|---|---|---|---|---|
| SAF | Southern Africa | 0.0, -8.1 | -0.4, -10.3 | 0.0, -6.6 | -0.1, -10.5 | 20.0, 95.9 |
| | | -13.0, -3.4 | -15.9, -4.4 | -10.7, -0.5 | -16.3, -2.0 | 77.9, 102.0 |
| TIB | Tibetan Plateau | 0.0, -0.8 | 0.0, -0.3 | 0.0, 0.4 | 0.0, 1.1 | 0.0, 14.3 |
| | | -0.7, -0.3 | -0.4, 0.4 | -0.3, 1.1 | -0.2, 1.6 | 9.3, 16.8 |
| WAF | West Africa | -4.5, -28.4 | -2.5, -21.8 | -5.6, -25.6 | -8.4, -26.5 | 175.3, 282.3 |
| | | -38.2, -20.4 | -29.7, -11.0 | -39.2, -10.3 | -44.0, -6.1 | 215.0, 392.1 |
| WAS | West Asia | 0.0, -2.6 | 0.0, -3.9 | 0.0, -4.4 | 0.0, -6.7 | 0.4, 24.8 |
| | | -3.4, -1.4 | -4.7, -2.5 | -5.2, -2.8 | -8.1, -4.6 | 18.3, 30.0 |
| WSA | West Coast South America | 0.0, -8.6 | 0.0, -10.5 | 0.0, -13.9 | 0.0, -21.2 | 57.2, 271.1 |
| | | -11.5, -5.5 | -14.5, -5.5 | -17.7, -7.6 | -25.1, -15.2 | 186.9, 346.3 |
| WNA | West North America | 0.0, 3.4 | 0.0, -3.5 | 0.0, 6.2 | 0.0, 6.8 | 23.5, 104.8 |
| | | 0.5, 5.6 | -0.1, 7.1 | 1.1, 11.6 | 1.7, 14.7 | 81.9, 126.7 |

## 3.2 Sources of ensemble variance

To investigate whether the main variance in projected GWR changes is caused by GHMs, GCMs, or the different RCP scenarios, we apply the Eq. (1) and (2) (see Sect. 2.4) for 1.5°C and 3°C GW. Figure 5 shows the GCM to GHM variance ratio
for 1.5°C (a) and 3°C (b) per grid cell; GHM RCP variance ratio is not shown here (see Fig. S7 in the supplement, mean of GHM RCP ratio: 22%) as the primary influence can be appropriated to the GCM and GHM selection (this is also the case when choosing only the $CO_2$ sensitive models). For the simulated variance at PI see Fig. S1 and S4.

Overall, GHMs cause more significant variance in 1.5°C than in a 3°C world, which is plausible because of increased GCM trends with increased $CO_2$ concentrations. Possibly this is also due to the missing RCP 8.5 simulations for PCR-
GLOBWB for all GCMs. A clear spatial pattern of GCM influence shows in the Amazon that relates to the region of Fig. 3 where increases of GWR are calculated. On the other hand, the region in the Amazon where decreases are simulated (compare Fig. 3) shows mainly the GHMs as the source of variance. In the Mediterranean, the influence shifts as well from GCMs (1.5°C) to GHMs (3°C). This could be due to a high agreement in GCMs in this region and a considerable disagreement in GHMs. Similar patterns can be found when comparing absolute GWR, but the influence of GCMs is less pronounced,
especially in the Amazon (Fig. S8).

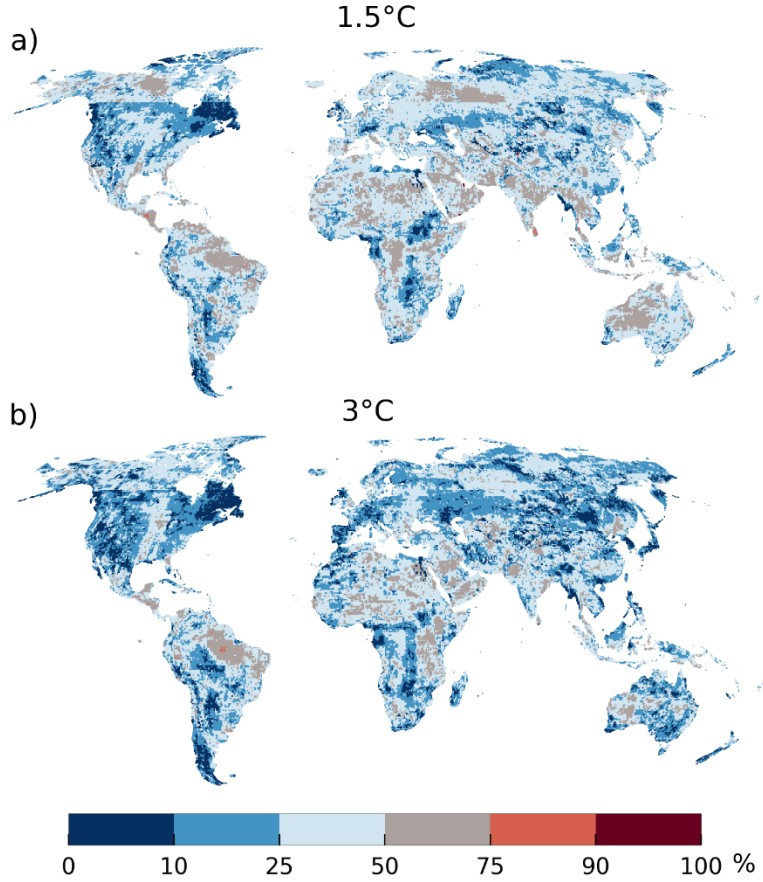

**Figure 5** GCM variance in percent of the total variance of GWR change from eight GHMs and four GCMs at 1.5°C (a) and a 3°C (b) GW (see also Sect. 2.4). Red depicts areas where the GCMs are responsible for the majority of the variance in GWR change. Blue areas indicate where the main variance is introduced through GHMs.

### 3.3 Impacts of evolving carbon dioxide concentrations on groundwater recharge estimates

Including vegetation dynamics in GHMs may alter the model response in future estimates of GWR as evolving $CO_2$ concentrations alters fluxes of energy and water (Davie et al., 2013). To investigate the influence of simulating the physiological impacts of evolving $CO_2$ on GWR, we compared GWR changes computed by two CLM 4.5 runs, each of it driven by GFDL-ESM2M climate input: the standard run analyzed included in the ensemble analysis above, with $CO_2$ concentrations changing according to the RCP, and an additional run in which $CO_2$ concentrations after 2005 were held constant at the 2005 level. Unfortunately, no other GHM-GCM combinations with these alternative $CO_2$ concentration variants are available in the framework of ISIMIP2b.

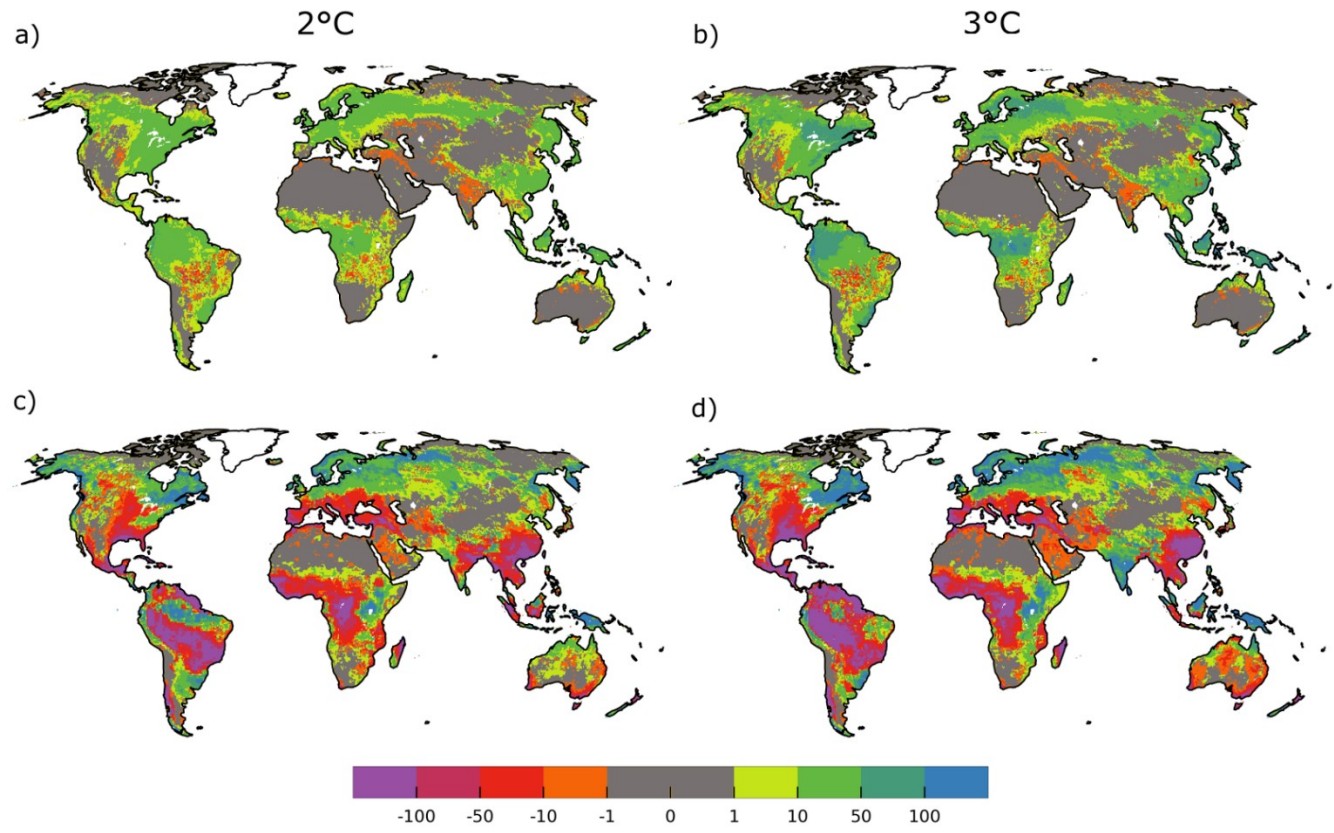

**Figure 6** GWR (dynamic $CO_2$) – GWR (static $CO_2$) [mm year$^{-1}$] for 2.0°C (a) and 3.0°C (b) GW. GWR (dynamic $CO_2$) – PI (dynamic $CO_2$) [mm year$^{-1}$] for 2.0°C (c) and 3.0°C (d) GW. The figure only includes the GHM CLM 4.5 and the GCM GFDL-ESM2M. Maps show changes in GWR at a certain GW (including all RCPs that lead to that GW with a certain $CO_2$ concentration) with dynamically evolving $CO_2$ compared to static $CO_2$ concentrations from 2005. Green and blue means that GWR is higher when evolving $CO_2$ concentrations are considered, red and purple less GWR.

Figure 6 shows differences in simulated GWR between a dynamic and a static $CO_2$ simulation for 2°C (Fig. 6a) and 3°C (Fig 6b). In most grid cells, GWR simulated with dynamic $CO_2$ is larger than GWR simulated with static $CO_2$ levels of 2005 (Fig. 6a,b). In the tropics, GWR with dynamic $CO_2$ can be higher than with constant $CO_2$ by 10-50 mm year$^{-1}$ for 2°C GW (Fig. 6a), while difference reaches 50-100 mm year$^{-1}$ in the 3°C world (Fig. 6b). Decreases of GWR are spatially consistent (for example, Brazil, Central U.S., and India) at 2° and 3°C GW and rarely exceed 10 mm year$^{-1}$.

Compared to the absolute changes between PI and the GW levels for dynamic $CO_2$ (Fig 6c,d) the decreases in GWR are rather small (e.g., up -10 mm year$^{-1}$ in Brazil (Fig. 6a,b), while change compared to PI exceeds -100 mm year$^{-1}$ (Fig 6c,d)). Also, increases in GWR due to dynamic $CO_2$ are in regions with large (> 100 mm year$^{-1}$, Fig 6c,d) increases in recharge.

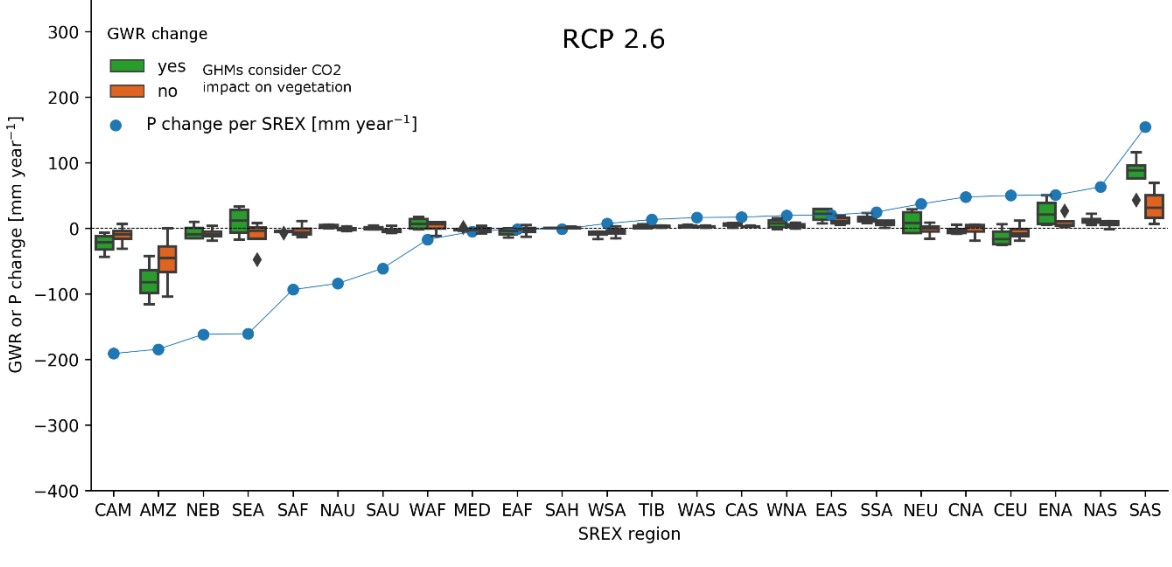

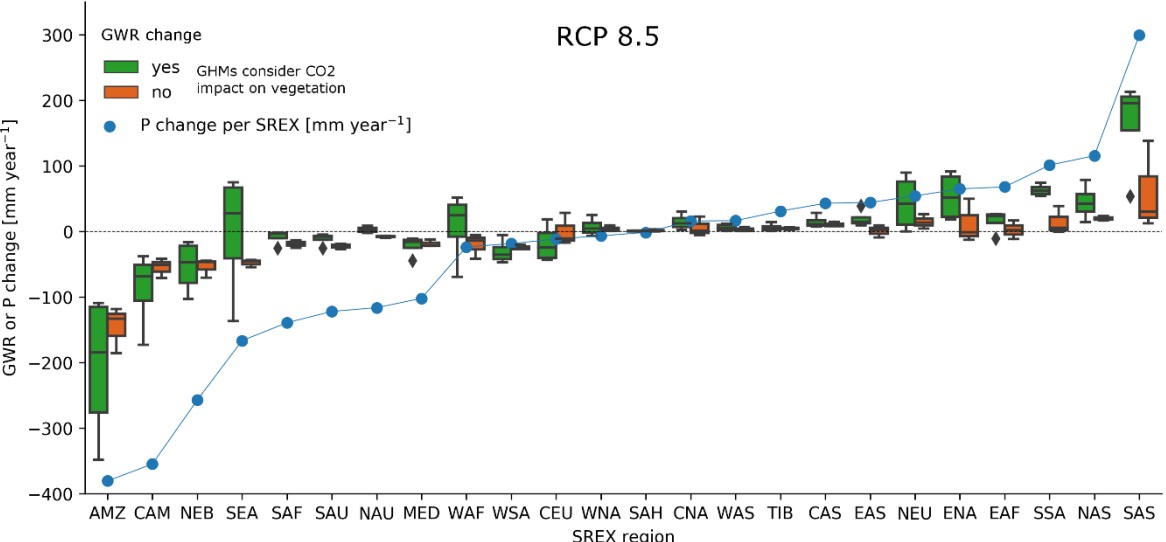

**Figure 7** Relation of changes in precipitation (P) (mean(1981-2010) – mean(2070-2099)) to changes in GWR (mean(1981-2010) – mean(2070-2099)) depending on the model type (with or without $CO_2$; see also Table 1) per SREX (selection as in Table 3) for RCP 2.6 435 and RCP 8.5 for the GCM HadGEM2-ES.

The preceding analysis focused on GW levels parallel to other studies of GHM ensembles. To investigate the difference in

including active vegetation processes in GHM further, we compared the four GHMs that include these processes with the four

models that do not (Table 1). Because different RCPs decide the concentration of $CO_2$ in the atmosphere, we compare RCP

2.6 and RCP 8.5 time slices instead of GW levels.

Figure 7 compares the precipitation and GWR changes between the period 1981-2010 and the period 2070-2099 for the two RCPs and the two different model types for the SREX regions investigated in Table 3. Changes in precipitation and GWR are only based on the GCM HadGEM2-ES (see Fig. S9 for average over all GCMs) as the relationship between GWR and precipitation is not linear and the plot is comparable to Davie et al. (2013), who investigated differences in runoff. Compared to the average precipitation of all GCMs where only two regions show a decrease larger than 100 mm year$^{-1}$ (Fig.
S9 (b)), HadGEM2-ES shows seven regions for RCP 8.5 with such a decrease in precipitation.

GWR changes vary between RCPs and model type and in between GHMs (Fig. S10). The relation between precipitation and GWR and difference between model types becomes clearer with RCP 8.5 than with RCP 2.6. Models with active vegetation (Fig. 7, green markers) agree that with more precipitation GWR should increase, e.g., for SAS; however, they disagree in regions where decreases in precipitation are expected and risk for groundwater availability is highest, e.g.,
CAM and MED. GHMs without active vegetation (Fig 7, orange markers), on the other hand, show a more consistent decrease in GWR for regions with decreases in precipitation and only some agreement in regions with increased precipitation.

Decreases in precipitation may lead to a decrease in vegetation productivity (if not counteracted by an increased water-use efficiency due to elevated $CO_2$ concentrations (Singh et al., 2020)) and thus to a decrease in transpiration. GHMs assume shares for evapotranspiration (ET) in relation to potential ET and the available precipitation. In contrast, transpiration
in $CO_2$-driven models responds to active vegetation as well as the relations between different water flux components that simpler GHMs do not. This can explain why the dynamic vegetation models exhibit inter-model regional differences in the GWR response to P decrease. Further, some models (MATSIRO) may not calculate LAI (leave area index), which impacts transpiration. For models with active vegetation, the increase in water use efficiency due to stomatal conductance (also referred to as $CO_2$ fertilization) can compensate for the decrease in precipitation to some extent, making more water available for
groundwater recharge as compared to the GHMs (Table 1). Though in some regions, as seen in Figure 7 (and Fig. S10), this feedback is not enough to overcome the warmer and drier climate in terms of groundwater flux. Overall, the capability of a model to simulate actual ET largely influences its capability to simulate groundwater recharge.

CWatM often lies in the middle of simulated GWR changes at RCP 2.6. Davie et al. (2013) showed generally higher runoff values for JULES-W1 than for LPJmL, the reverse is true for GWR (Fig S.10). For RCP 8.5, CWatM always simulates
the largest increases and lowest decreases in GWR of all models without active vegetation.

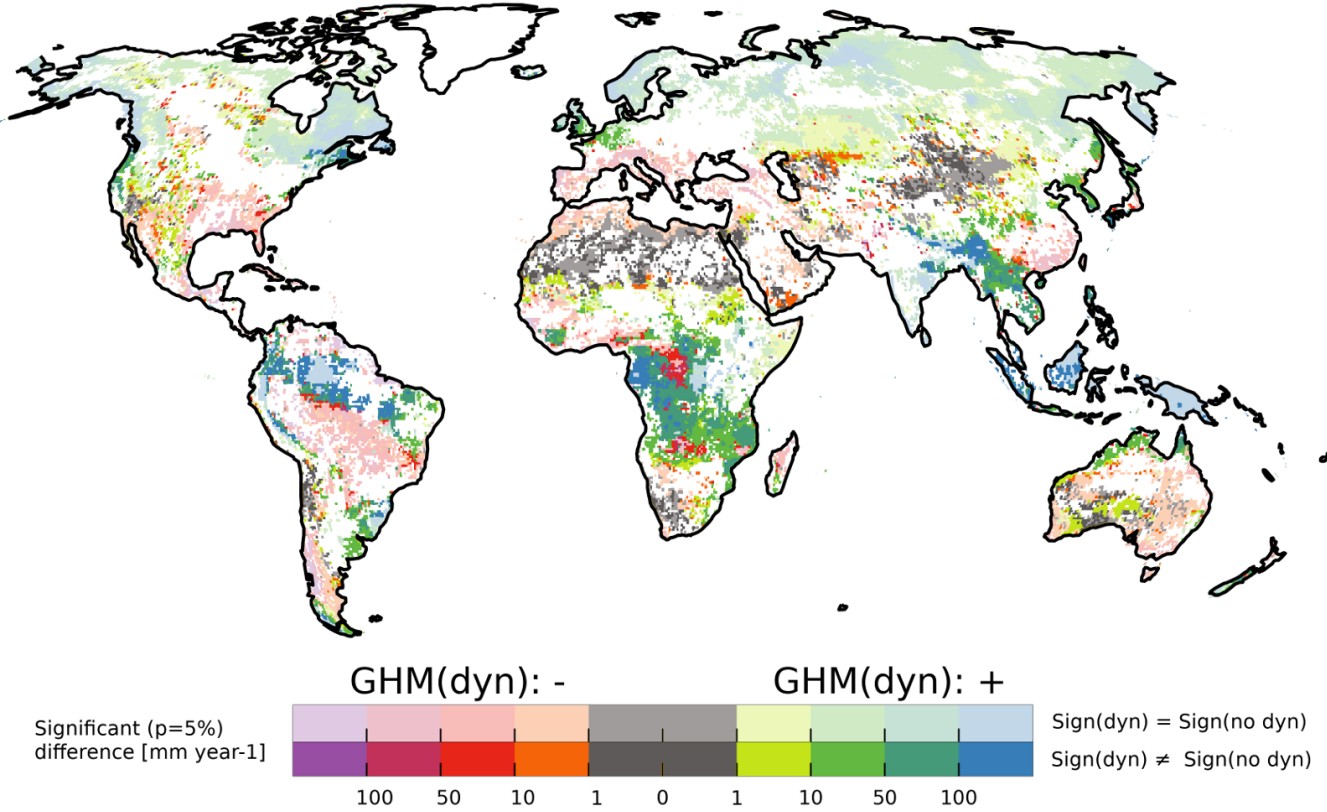

**Figure 8** Significant absolute difference of GWR change between 1981-2010 and 2077-2099 for RCP 8.5 in between four GHMs with (dyn) and four GHMs without dynamic (or active) vegetation (no dyn). See also Table 1. Reddish (left side of the color bar) indicates that the mean change of GWR as computed by the models with dynamic vegetation is more negative or less positive than change computed by other models. White regions indicate no significance is based on the K-S test (Sect. 2.5). Solid colors indicate that the majority of the two model groups (3 out of 4 models for each group) do not have the same sign i.e. that including dynamic vegetation leads to different signs in GWR change. Lighter colors indicate where the majority agrees on the sign of change.

A spatially more refined difference between the model types is shown in Fig. 8 for RCP 8.5 (For RCP 2.6, almost no significant changes were found). For each grid cell, the map shows the significant (K-S test, p=5%) absolute difference of simulated change in GWR between models that include dynamic vegetation processes and models that do not include them. In the northern latitudes, both models with and without dynamic vegetation agree on an increase in GWR but differ by up to 100 mm year$^{-1}$. Similarly, in the Mediterranean and central Brazil, both model types simulate a decrease in GWR, but the magnitude is significantly different between the model groups. In the Amazon patches of significant differences between the models show increases of GWR computed by GHMs with dynamic vegetation, whereas GHMs without dynamic vegetation shows a decrease. A similar effect is visible in central Africa, India, and parts of Indonesia; however, also decreases are simulated instead of increases for the Congo and Zambesi catchment. Both in the Mediterranean and South America models with dynamic vegetation show up to 100 mm year$^{-1}$ difference in change compared to models without, even though no physiological effect should be dominant. According to Fig. 6, this is likely due to CLM 4.5 because JULES-W1 and LPJmL show slighter GWR

decrease than the models without dynamic vegetation. It is likely that the shown differences are due to the implementation of dynamic vegetation in the GHMs (compare Fig. S.10), however it is possible that other model peculiarities and processes are relevant as well.

## 4 Discussion

Estimating GWR is challenging (Moeck et al., 2016). Our results show that even for the PI period, the estimates of GWR vary largely among different GHMs. This is likely caused by the very different treatment of the runoff partitioning, implementation of the soil layer(s), inclusion of dynamic vegetation processes, and simulation of capillary rise. Because GWR is hard to measure directly (Scanlon et al., 2002), it is also challenging to verify the accuracy of the estimates.

To the best of our knowledge, the data-set of Mohan et al. (2018) is the only available gridded global GWR dataset that is not based on global hydrological modeling. This data set of mean 1981-2010 GWR in 0.5° grid cells was developed from a regression analysis that combined gridded datasets of mean precipitation and potential evapotranspiration as well as land use/land cover with local estimates of GWR at 715 locations worldwide. Figure 9 compares the GHMs under investigation for PI conditions to this dataset. The used data for comparison is one ensemble member of the analysis of Mohan et al. (2018) that was deemed best in their study. The global mean GWR in this member is slightly lower, 110 mm year$^{-1}$, than the reported mean of 134 mm year$^{-1}$. Overall, the GHMs best agree with Mohan et al. (2018) in arid regions like the Sahara, Australia, southern Africa, and the Andes. Underestimates are predominant in the northern Latitudes and Central Asia, whereas underestimates appear in Europe and the eastern USA for all models. All models, except for H08 and WaterGAP2, which show underestimates, result in overestimates in East Asia. In the Amazon, MATSIRO and CLM 4.5 overestimate by more than 100 mm year$^{-1}$ compared to Mohan et al. (2018), whereas all other models show a mix of over and underestimate across continents. A similar pattern is visible in Central Africa where CLM, MATSIRO, and CWatM overestimate, and all other models show a mixture of over and underestimate of -100 – 100 mm year$^{-1}$. H08 and WaterGAP2 have the best agreement according to the NSE (Nash-Sutcliff Efficiency (calculated spatially); (Nash and Sutcliffe, 1970)) of 0.4 and 0.2 while the mean bias (mean(GHM Mohan et al.$^{-1}$)) is lowest for JULES-W1. All GHMs show much lower GWR in permafrost regions as they assume that there is no or little GWR in such regions. Possibly GWR of Mohan et al. (2018) is overestimated here as no measurements informed their results in these regions.

The variance in modeled GWR is possibly caused by the different implementation of the hydrological processes in between the models. Even more, models differ in their definition of groundwater and GWR. Some include groundwater storage that is recharged by a fraction of precipitation others do not include a groundwater component at all but define the saturation excess water from the bottom soil layer as GWR. Models may include only some of the processes that affect GWR, for example, capillary rise, percolation from the soil, preferential flow bypassing the soil matrix, the interaction between surface water and the aquifer, changing land use over time (not considered here), changing vegetation (e.g., reducing infiltration capacity). Further, important processes like evaporation, infiltration, percolation, or runoff and GWR separation are

implemented with different equations and simplifications. For evapotranspiration, a standard deviation of 0.15 mm day$^{-1}$ globally for the period 1989–2005 was found in the ISIMIP ensemble (Wartenburger et al., 2018). Some models even use sub-grid information or sub-daily time steps, e.g., for changes in unsaturated conductivity. Notably, models that include dynamic vegetation processes showed the largest spread in GWR in regions with decreasing precipitation. It is also important to distinguish the capability of models to computed groundwater recharge during a historical period from their capability to estimate changes of groundwater recharge due to climate change. A model that simulates the current groundwater recharge pattern correctly may be incapable of computing future groundwater recharge if it cannot correctly simulate the impact of climate change and changing atmospheric $CO_2$ concentrations on actual evapotranspiration correctly.

To illustrate the model differences further, the following describes the impact of changes in precipitation for WaterGAP and LPJmL representative for the different model types used in this study. In WaterGAP, a simulated percent change in total runoff translates to the same percentage change in GWR; unless, e.g., due to more extreme precipitation events, infiltration capacity is exceeded more often such that the relative increase in GWR is smaller than total runoff. Absolute changes in GWR are always smaller than changes in total runoff. In LPJmL, changes in total runoff do not translate to proportional changes in groundwater runoff and GWR. Any flux or storage that takes water before it is partitioned to the soil will impact the groundwater and GWR. Possible reasons for a reduction in GWR (percolation past the bottom hydrologically active layer (3 m deep); compare Sect. 2.1) can be changes in precipitation amount/intensity, transpiration due to vegetation productivity, transpiration due to changes in vegetation water use efficiency due to $CO_2$ fertilization, or changes in anthropogenic water use demands.

This difference in behavior is reflected in Fig. 7, where the response between precipitation and GWR of GHMs without any active/dynamic vegetation is relatively uniform. The non-uniform response of the models that include vegetation changes is likely due to the complicated process feedbacks between vegetation and water (transpiration changes due to available water together with vegetation productivity) and complex feedbacks in-between changes in $CO_2$, temperature, and precipitation which affect vegetation.

This study highlights that uncertainties and differences in GHMs need to be investigated further and that in order to estimate global groundwater vulnerability, improved estimates of global GWR are required.

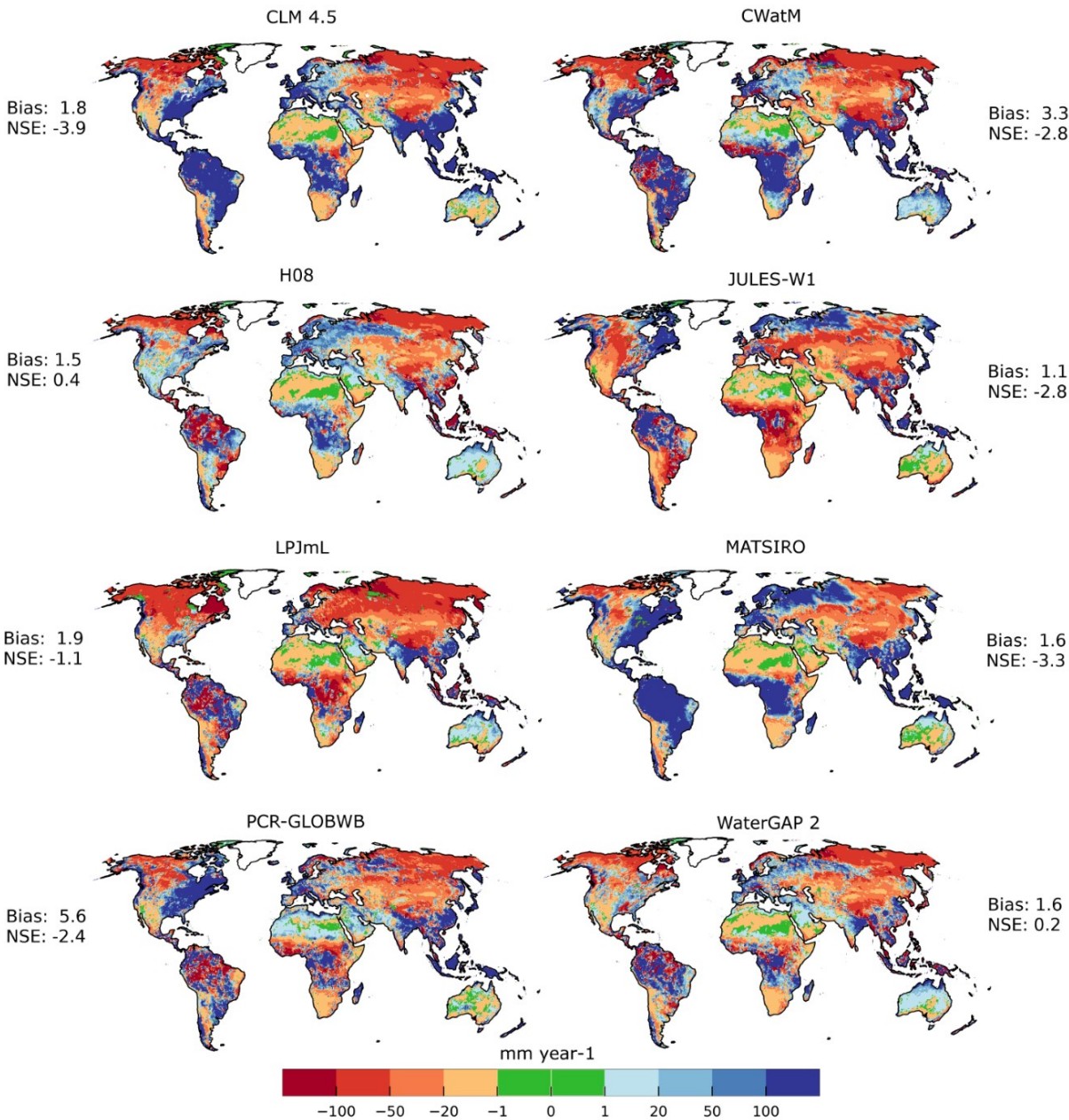

**Figure 9** PI GWR per GHM – 34 years (1981-2014) mean GWR [mm year⁻¹] of Mohan et al. (2018). Bias: mean (GHM Mohan et al.⁻¹). NSE (Nash-Sutcliff Efficiency; (Nash and Sutcliffe, 1970)) is calculated spatially over all cells instead of time.

This study is limited not only by the uncertainty in correctly representing the process of GWR but also in the propagation and aggregation of uncertainties. Future greenhouse gas emission scenarios are created based on the input of integrated assessment models. They are translated into emission scenarios of atmospheric concentrations and forcings that are, in turn, used to evaluate their impacts on the climate simulated by GCMs. Outputs of the GCMs are then bias-adjusted and

spatially downscaled to be used in the assessment with impact models like GHMs (Döll et al., 2014a). Furthermore, the analysis is limited by the number of GCMs that were used, as discussed in McSweeney and Jones (2016). Although the GCMs are 550 carefully selected to be most representative of the CMIP (Taylor et al., 2012) ensemble.

The multi-model ensemble study presented here assesses GWR at GW of 1.5°C, 2°C, and 3°C compared to GWR simulated under pre-industrial climate conditions and 1°C of GW. Changes are assessed based on transient time slices of the 30 years around the year that crosses the specific warming level. These slices are an approximation of the stabilized climate state of that warming level; it relies on the assumption that for a given warming level the impacts are the same regardless of 555 the time it took to reach it or whether equilibrium has been reached at all (Boulange et al., 2018). However, this kind of analysis has limitations as the transient nature of climate is aggregated over a relatively short period (31 years). Components like the ocean might not equilibrate at these timescales (Donnelly et al., 2017).

Additionally, different RCPs are combined, which limits the possibility to investigate processes that are sensitive to different $CO_2$ concentrations. Investigations in this study based on RCPs show the difference between these model types. On 560 the other hand, using GW levels reduces the uncertainties from GCM variability due to the use of different time slices, depending on when a GCM reaches a GW level.

The variance in GWR is caused by GCMs and GHMs alike depending on the region similar to a multi-model ensemble study on the climate change impacts on streamflow (Schewe et al., 2014). Again, the assessment is limited by the number of used GCMs. Furthermore, this study did not include changes in land-cover and land-use, and thus irrigation which 565 can have a tremendous impact on GWR, especially as irrigation patterns and used crops, will change with a changing climate (Hauser et al., 2019; Hirsch et al., 2017; Hirsch et al., 2018; Thiery et al., 2017; Thiery et al., 2020).The only similar study on the global impacts of GW on GWR, to the knowledge of the authors, was conducted by Portmann et al. (2013). The study used five GCMs and one GHM, WaterGAP, which (a slightly different version) was also included in this study. Overall results are spatially consistent; however, Portmann et al. (2013) showed more consistent trends among GW levels (compare Table 3). 570 Portmann et al. (2013) acknowledge that including impacts of evolving $CO_2$ levels on vegetation will have an impact on the simulated GWR and that WaterGAP is likely overestimating the decreases in GWR. Similarly, Davie et al. (2013) found that simulation of runoff was not consistent across models depending on whether $CO_2$ was considered. The results presented in this study show that this assumption is true for some regions, where differences of up to 100 mm year$^{-1}$ can be observed.

Despite the uncertainties, this study provides further evidence that climate change will impact groundwater 575 availability in many regions of the world. A notable decrease can be expected in the Mediterranean, Amazon, and Brazil, whereas increases can be expected in Northern Europe. It is nevertheless troublesome that, especially in regions that are known to be vulnerable to climate change, for example, South Africa, model agreement in between model types is that low.

## 5 Conclusions

Potential GWR changes due to climate change require increased attention from the scientific community as well as from decision-makers because they affect future water availability in many regions and thus the wellbeing of billions of people. This study shows that simulated global-scale estimates of GWR vary strongly among GHMs, which contribute more strongly to the overall uncertainty of future GWR than the applied GCM output. However, statistically significant increases and decreases of GWR could be identified in specific regions per GW level. The presented inter-model ranges of GWR changes are an important input for processes aiming at developing strategies for climate change adaptation, as risk-averse decision-makers may want to orient their strategies towards adapting to the worst-case GWR change and not to the projected ensemble mean change.

This study shows that including vegetation processes in GHMs can change projected GWR changes substantially. However, consideration of these processes does not lead to a uniform increase of groundwater recharge, as might be expected from the physiological effect of increasing atmospheric $CO_2$ concentration. In some regions with decreasing groundwater recharge, where groundwater availability is a major concern, models that include these processes show the largest differences among themselves. Further research is necessary to understand GWR on large scales, and how it is affected by climate. Simulation of groundwater recharge in global models and the connected uncertainties need to be analyzed in greater detail by, e.g, the application of extensive sensitivity analysis. Such an assessment should also extend to the benefit of integrating gradient-based groundwater flow models in GHMs.

## Data availability

All simulations are available through the ISMIP project at https://www.isimip.org.

## Acknowledgments

We like to thank the ISMIP (https://www.isimip.org) project for supplying the data and the modeling community for carrying out these crucial simulations. We furthermore like to thank Chinchu Mohan for providing the data. This research has been supported by the German Federal Ministry of Education and Research (BMBF, grant no. 01LS1711F).

## Author contributions

RR led the conceptualization, formal analysis, methodology, software, visualization, and writing of the draft—original idea by HMS. HMS and PD supported review and editing, as well as the development of the methodology. TT supported editing and review. MF, SNG, MG, NH, AK, LS, WT, YW, YP, BP, and SY contributed to the model description in section 2 and made

suggestions regarding wording, figures, and discussion. PD and HMS supervised the work of RR and made suggestions regarding the analysis, structure, and wording of the text and design of tables and figures.

**Competing interests**

No competing interests.

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
