# Peer review of "Uncertainty of simulated groundwater recharge at different global warming levels: A global-scale multi-model ensemble study"

_Hydrology and Earth System Sciences, 2020_

## Referee Comment (RC1) · Anonymous Referee #1 · 2 Jul 2020

This study is impressive in the number of global models included (8) driven by four global circulation models considering three representative concentration pathways, totaling 86 different cases. Although the title of the paper and the results emphasize model uncertainties, I think the primary result based on the analysis is that recharge is not very sensitive to climate change as only 15% of cells show significant increases or decreases in recharge based on pre-industrial baseline and only 8% of cells show significant change in recharge from current 1 degree to projected 3 degree condition. It would be good to acknowledge that recharge is likely the most difficult component of the water budget to simulate because it is modeled as a residual, accumulating uncertainties in other water budget components. In addition, it is extremely difficult to

simulate in semiarid regions because small uncertainties in precipitation and evapotranspiration can result in large uncertainties in recharge. Many studies suggest that climate change will result in increased climate extremes (floods and droughts) that may result in increased recharge from focused rather than diffuse recharge; however, it seems that few of the models consider focused recharge. The authors refer to groundwater levels throughout the paper with respect to temperature levels; however, this is confusing as groundwater levels are generally considered water table levels. It might be good to include temperature when referring to these. I agree that it is good to focus on absolute changes in recharge rather than relative changes. The authors suggest that underestimating runoff would result in increased GWR; however, this would not be the case if GWR is focused and derived from runoff as in semiarid regions (L. 74). The authors repeatedly use present tense to refer to work that was done for this study. I think it would be more appropriate to use past tense. The model CLM-5 has been upgraded substantially relative to CLM-4.5. It might be good to consider CLM-5 rather than CLM-4.5. Soil thickness varies substantially among the models (e.g. LPJ 13 m thick). It would be good to comment on the impact of varying soil thickness on model results.

---

## Referee Comment (RC2) · Anonymous Referee #2 · 5 Jul 2020

The study presented by Reinecke et al. is based on a very important set of modeling, a very impressive work. The authors gather 8 Global Hydrological Model (GHM), 4 Global Climate Model (GCM),and 3 emission scenarios RCP. They address the uncertainty on the simulated groundwater recharge (GWR). Especially, an effort is made to estimate the uncertainty associated to the explicit account of dynamic vegetation. Especially, it was nice to aggregate the results according to global warming (GW) thresholds.

If the manuscript is of great interest, some parts of the analyzes are not clear. One of the main conclusions is that dynamic vegetation has a strong impact on the estimated GWR, which is far from being as obvious based on the results presented ... In addition,

some figures are difficult to read and seem to support only partially the comments. You'll find below my mains remarks.

- Introduction: The choice is made to estimate ground water recharge throughout the continental part of the world. However, many areas have no extended groundwater. Whymap.org provides map of the extension of the aquifer. What does it mean to estimate GWR where there is no or local and shallows aquifer? Won't it be more interested to estimate the recharge on the aquifer domains?

- Line 200: It is necessary to provide information on the bias-adjusted method? What is the assumption? What is the reference climate used ?

- Section 2.1: I suggest to provide a table that summarizes if GHM includes or not in the GWR another part that a partition of the precipitation, especially, which GHM includes river to aquifer exchange, since this may be a very important difference in alluvial regions. Moreover, this table should also summarize which GHM integrates direct effect of $CO_2$, explaining clearly if they account for stomatal aperture sensitivity to $CO_2$ and/or for vegetation dynamic (LAI) (so far, this is not clear).

- Figure 2 is of bad quality... Is it necessary to have all the extremes? Is it reasonable to have a range over 2000mm/year? most aquifers recharge maps stop before 1000mm/year and often under 500. Are GWR values above 1500 in Figure 2 located in capacitive aquifers or in very local and shallow aquifer as defined by Whymap (see comment 1)? As this figure is difficult to read, it is impossible to check comments line 317-322

- Nice to disentangle the impact of GCM and GHM, but, fig3a includes 76 cases while fig b includes only 36 cases. How does this compare? In order to try to understand what the impact of GHM is, and what the impact of the response

[Figure]

of GHM to GW is, it seems required to show the variance of the 8 GHMs on preindustrial case. Figure 7 shows that there are important change, but, variance will be helpful to compare with fig3a.

- Figure 4 includes only one realization of one GHM and one GCM.... Why this GCM? Why this GHM? Is this GCM includes dynamic vegetation ? Is the dynamic vegetation of the GCM consistent with the one of the GHM? In any case, it would be nice to have some information on LAI changes...;

- Line 422 : Decrease of precipitation lead to a decrease of vegetation productivity ⇒ I guess it is more complicated than that, you may correct

- Figure 5: is this figure correct? a same GHM can appear twice for a subregion with the same colors... A change of precipitation of 100mm/year can lead to an increase of GWR of the same amount... Is the vegetation dead ? It would be nice to better understand which process occurs in detail in one subregion ?

- Fig 6 compares 4 GHM with dynamic vegetation (change in LAI?) with the 4 others who assumes constant vegetation (but do they account sensitivity of the stomatal aperture to CO2?). The difference on the physics of the GHMs is large, but the impact seems reduced. How can we be sure that the difference is linked to the dynamic vegetation? Similar difference may well exist between this different GHMs without change in vegetation dynamic... Moreover, are the significant changes located where the aquifers are extended and capacitive, or where the aquifers are very shallow and local?

- Assessment of the GHM should not appear in the discussion.... It should be earlier, or in supplement... GWR estimates by Mohan is only the GWR from precipitation (no river inputs). How does this compare to the numerous GHM ?

- Line 539: "Despite the uncertainties, this study shows that climate change will

impact groundwater availability in many regions of the world".: this is naïve: it was already shown by numerous regional studies. . ..

- Line 545: "Moreover, this study shows that including dynamic vegetation processes in GHMs can change the results substantially": this is not that clear from your results. . .. I was expected more impact indeed. . .

Other remarks:

- A map of the extension of the subdomain is required

- Numbering of figures is wrong

- Line 287: Masson-Delmotte et al. . .

- Figure S3 should be in the text since it is discussed

---

## Referee Comment (RC3) · Anonymous Referee #3 · 14 Jul 2020

Reinecke et al. study the influence of different global warming levels on groundwater recharge using a suite of hydrological models at the global scale. They report that uncertainty in the GWR estimates is large, and may be possible with confidence in specific regions.

I do appreciate the tremendous amount of work the authors put into the study and already apologize for not being able to be more positive. The study has a flaw in that it assumes/postulates that GHMs are able to simulate groundwater recharge processes (line 61). This remains to be shown (see specific comment below). The authors also compare to Mohan et al. (2018), a data set, which is also highly uncertainty itself,

and show that all models essentially have no skill. Thus, the study is hypothetical and should be seen as a model sensitivity study, which does not necessarily reflect reality. The large uncertainty in the results supports that notion. The authors discuss the limitations in detail and come to the conclusion (line 538) "Despite the uncertainties,this study shows that climate change will impact groundwater availability in many regions of the world." Yes, that's probably true. But it's sad that due to the large uncertainty, no additional concrete conclusions can be drawn from the results.

Because the detailed analyses and numbers have very low confidence, I am not able to comment on the simulation results.

In summary, I find that GHMs have not been tested comprehensively especially with regard to complex processes such as groundwater recharge. Thus, in my opinion, the study is too early (first testing, then analyses). Looking at the description of the models, I would not even call the estimated flux groundwater recharge. If GHMs are applied I suggest to study the major fluxes of the water balance including their impact on the residuals of the balance equation in the models. In a way that's what the authors are doing, yet the presentation does not show these results.

Only a couple specific comments Abstract: The reported percent increases/decreases of GWR suggest an accuracy that is simply not there; especially given the huge uncertainty in the results. Thus, the abstract sends the wrong message, especially to water managers and decision makers.

54: One of the most important factors is missing: Depth and dynamics of the free water table.

61: This is true, but has never been tested. I suggest to compare the GHMs against fully integrated hydrologic models, such as Cathy, Hydrogeosphere, OpenGeosys, ParFlow, etc. in order to test the ability of GHMs to simulate recharge processes. This is one of many tests that GHMs should undergo in my opinion.

Section 2.2: Porbably I missed these details: what's the time step, the spatial resolution, etc?

---

## Author Response (AR3)

Dear Philippe Ackerer,

Thank you very much for your efforts in finding reviewers for this manuscript and the opportunity to revise it. Even though both reviewers certainly added interesting remarks on how to improve the paper, we are concerned that most of their suggestions are out of scope or would alter the paper to an extent that would significantly change the paper's character.

With this, we would also like to respond to your direct suggestions of:

(1) improve the discussion of some results (sometimes, it is too descriptive)

> *We assume that this request is linked to the reviewer comments to explain more deeply how the parametrisation of the models is linked to the responses in groundwater recharge and that it should be compared to other modelling efforts. As explained below, this entails a very complex and extensive analysis of the models going well beyond what we have intended for this paper. However, we now state more clearly where the model setup parameters can be found and discuss issues related to estimations of ET.*

(2) provide an insight on model reliability

> *In our manuscript model reliability, regarding the estimation of global-scale groundwater recharge, was evaluated by comparison to Mohan et al.. We agree that this can only be a first step into investigating the model reliability. However, providing insights on model reliability is challenging, which entails a better understanding of how uncertainties propagate through these complex models. As we discuss in more detail in the replies, we still lack the methods of, e.g., applying extensive sensitivity methods to GHMs to further investigate these issues. Some preceding work, e.g., Döll and Fiedler (2008), have been carried out to investigate simulated recharge reliability by using expert knowledge and some regional studies. This work is cited in our manuscript, but conducting additional experiments on this would go significantly beyond this paper's scope, which focusing on the differences among the models in the ISIMIP2 framework and the influence of $CO_2$ changes on recharge.*

> *Also, investigating the reliability regarding future climate change-driven groundwater recharge changes cannot be investigated by comparison to observation data. We assume in our study, as it is usually done in multi-model climate change assessments, that all models are considered to be equally reliable.*

(3) provide some model parameters for some region.

> *We now explain more clearly that our study has been conducted in the framework of an established protocol (ISIMIP) and where to find more information about the models' parameterisation. A regional discussion (also for only one region) of parameters would require a different study setup that would require the concerted work of multiple model development teams over an extended timeframe, delaying the publication by years. Due to the complexity of daily simulations with multiple complex hydrological models (and different climate scenarios), truly knowledge-based statements about sensitivities in specific regions go well beyond the paper's scope.*

We want to emphasise that our rebuttal does not originate from an unwillingness to accept the reviewers' well-founded concerns but from our conviction of what is possible and what we deemed to be the core messages of the manuscript: (1) the inclusion of vegetation processes can substantially change the projection of groundwater recharge changes, (2) the estimation of recharge varies largely in-between models and requires further investigation, and that while (3) taking into account the model disagreements statistical significant changes of global groundwater recharge can be observed for specific regions under different global warming levels.

The following lists all comments of the two reviewers and our rebuttal in *italics*. Attached is a markup document that highlights the changes compared to the last submitted revision. Line numbers refer to the revised version of the document.

**#1**

**1.1**

3 reviewers have provided their insights in the first round. I strongly agree with reviewer 3 in the sense that the study is at its early stage and there are several things to be addressed. The reviewer raised some important and key points that the authors missed to address. I will advise the authors to address these issues (e.g. the use of groundwater recharge and the large uncertainties of their results). The authors would rather focus on fluxes and water storages which is what these models are actually simulating or clarify that they are showing an effective recharge, not the actual recharge, which is much more complex.

*We agree that our work shows that global groundwater recharge in GHMs is still uncertain and needs to be investigated further. This is one of the main contributions of the paper and clearly stated.*

*Regarding the precise definition of the term "groundwater recharge", none of the models simulates the depth of the groundwater table beneath the land surface. Hoping to understand the reviewer correctly, we would agree that our models do not compute the actual timing of the groundwater recharges and in that sense not the "actual groundwater recharge", in particular, if the groundwater table is very deep. Therefore, the groundwater recharge response at the location of the groundwater table to climate change may be delayed in case of deep groundwater table occurring in particular in dry regions of the globe. We now explain this caveat in section 2.1:*

*[132 ff]: "We do not consider focused recharge in this study as no model offers a reliable implementation of these processes until now. Also, none of the models simulate he depth of the groundwater table beneath the land surface which does not allow to correctly attribute delays in recharge due to water table depth."*

*Overall we are convinced that it is time to do a study that aims to understand the best information we have on potential impacts of climate change on groundwater at the global scale, which is provided by the multi-model ensemble output analysed in this study. Our study's merit is similar to many climate change studies done with global climate models used in combination to impact models (e.g. global hydrological models to study climate change impacts on streamflow or global crop models to study impact of climate change on yield). It is well known that uncertainty of groundwater recharge estimates are high, in particular at the global scale, and that different global climate models project very different*

*future climatic changes in response to the same greenhouse gas emissions scenario; nevertheless, model-based assessments including an understanding of their uncertainties are of interest to many and inform decision-making regarding climate change mitigation and adaptation. We explicitly wanted to focus on uncertainties and not (impossible) predictions, which is also reflected by our manuscripts' title: "Uncertainty of simulated groundwater recharge …". The manuscript is also very clear that the implementation of recharge varies greatly in-between the models and may or may not include different specific recharge processes. Our study goes well beyond the current state-of-the-art (e.g. Döll and Fiedler, Swenson et al. (2015) or Portmann et al. 2013). Unlike Portmann et al. (2013), where only one global hydrological model was applied, this study is the first to include the uncertainties stemming from different global hydrological models and how they simulate evapotranspiration and thus also groundwater recharge.*

**1.2**

Reviewer 3 also suggested comparing GHMs against fully integrated hydrologic models that simulate recharge processes, but the authors failed to do so. It would be great to compare in some areas how their models performed compared to the integrated hydrologic models. These models are now run on a continental scale. The model outputs could also be compared to some global datasets to ensure that they are at least consistent with observations at this scale and resolution.

*Parallel to our last response, we agree that such a comparison would be tremendous and should be targeted in future studies. However, implementing such a comparison in the current manuscript would go greatly beyond the paper's scope because a comparative framework (in terms of input data and modelling protocol) would be required. It is also essential to distinguish the capability of models to computed groundwater recharge during a historical time span from their capability to estimate changes of groundwater recharge due to climate change. The latter would be required to "validate" our study and is much more challenging. A future study might combine historical observations of groundwater recharge, integrated models and GHMs as driven by observational input data. We have added this to the outlook.*

*We have added the following text to the Discussion (after comparison to the independent global-scale estimate of Mohan et al.*

*[522]: "It is also important to distinguish the capability of models to computed groundwater recharge during a historical period from their capability to estimate changes of groundwater recharge due to climate change. A model that simulates the current groundwater recharge pattern correctly may be incapable of computing future groundwater recharge if it cannot correctly simulate the impact of climate change and changing atmospheric $CO_2$ concentrations on actual evapotranspiration correctly".*

**1.3**

The topic of this paper is relevant to the community. The authors mostly presented a bunch of results without explaining in detail why we are observing the changes and what is driving these changes, the most expecting part of this kind of study. In addition, the authors miss the opportunity to discuss the

setting up of their models (initial conditions, parameterisation, etc.), model validation is also an important step in modeling.

*Each of the presented models is very complex and describing the parameterisation, validation, initial conditions etc. of only one model is challenging. For example, see the most up to date descriptions of WaterGAP ([https://gmd.copernicus.org/preprints/gmd-2020-225/](https://gmd.copernicus.org/preprints/gmd-2020-225/) ) and PCR ([https://gmd.copernicus.org/articles/11/2429/2018/](https://gmd.copernicus.org/articles/11/2429/2018/)). Summarising these descriptions of all models go well beyond the scope of this paper. We have cited all relevant publications that describe the models and their setup in the paper and summarised the implementation of the process we are focusing on (recharge). Everything beyond that is a review paper with a different focus.*

*A paper that strives to summarise the model structure and parameterisation of all models that are considered in the study is now in review (*Telteu et al., 2021*). We have added this citation and clarified the parameterisation of the models. See also #1.7 for the altered text and #1.4 for an explanation regarding the drivers of change.*

**1.4**

Understanding what is driving the processes and the observed changes should be the key output of this study, this will advance not only our knowledge on the uncertainties associated with the simulated groundwater responses to climate change but also how can we reduce these uncertainties.

*By assessing to what degree the projected groundwater changes depend on whether the global hydrological models take into account process related to an active vegetation (e.g. closing of stomata and/or at higher atmospheric CO2) is an attempt to understand the drivers of changed evapotranspiration and thus groundwater recharge. We agree that we need a more detailed understanding of the uncertainties and what processes contribute to them. A further attempt for a better understanding is the already mentioned model review of Telteu et al. (2021) that enables insights into the process representations of those models. Future work will need to extend the review of Telteu et al (2021) and the data analysis as shown here by applying extensive sensitivity methods even though this is a very challenging task and demands for new methods that currently do not exist for these complex models. This is now more clearly reflected in the conclusions:*

*[Last line] "Simulation of groundwater recharge in global models and the connected uncertainties need to be analysed in greater detail by, e.g., the application of extensive sensitivity analysis."*

**1.5**

The authors have done tremendous work to develop such a global modeling framework. I would recommend the authors to thoroughly revise their paper, discuss the use of groundwater recharge, provide some explanations in the uncertainties they are observing, and compare these uncertainties to the ones associated with evapotranspiration.

*Thank you for the encouraging comment. We need to state that the modeling framework is not specifically dedicated to compare simulated groundwater recharge but to assess the impact of climate change on a large number of hydrological variables such a total runoff or floods. This study is the first*

*impact multi-model assessment of groundwater recharge from the "Inter-Sectoral Impact Model Intercomparison Project" (ISIMIP, www.isimip.org). The setup of the framework is discussed in described in* Frieler et al. (2018): *https://gmd.copernicus.org/articles/10/4321/2017/ as cited in the manuscript.*

*Regarding the term groundwater recharge, please refer to our answer to 1.1. Explanations of the uncertainties and their possible explanations are thoroughly laid out in the discussion section of the manuscript. To address this comment, we have further extended the discussion which now also compares the described uncertainties to the ones associated with evapotranspiration (see also* Wartenburger *et al* (2018*); https://iopscience.iop.org/article/10.1088/1748-9326/aac4bb for and extended discussion of evapotranspiration in ISIMIP).*

*[517 now reads]* "Further, important processes like evaporation, infiltration, percolation, or runoff and GWR separation are implemented with different equations and simplifications. For evapotranspiration, a standard deviation of 0.15 mm day$^{-1}$ globally for 1989–2005 was found in the ISIMIP ensemble (Wartenburger et al., 2018)."

**1.6**

1. The table added in the revised manuscript is very helpful. Nonetheless, I will make the table clearer without text. The authors should also clearly discuss the differences in the processes resolved by these models.

*It is unclear what the referee refers to here. Would the table be clearer with or without text? And is that related to the text within the table or the text that describes the table? The differences in implementation of the processes (We assume the referee refers to vegetation and recharge processes) are presented in the discussion section of the manuscript. We, however, recognise that a more extensive investigation in the uncertainties is merited. We state that now more clearly in the conclusions. See also #1.4 and #1.1.*

**1.7**

2. The authors should dedicate a section to discuss the types of data they use to build their models. Did they use the same datasets for all these models? How did they initialise their models? Before jumping into model comparisons, one needs to clearly understand the differences in the model parameters and initialisations. The authors discuss the climate simulations but since their paper is focused on groundwater recharge, it is the groundwater models and their uncertainties in computing the recharge that needs to be discussed.

*Again, we need to refer to the setup of the study clearly explained in the introduction of the paper that states that this assessment was conducted in the framework of the ISIMIP intercomparison protocol. It is likely that the referee assumes that this ensemble was only setup to compare groundwater recharge. See also our reply in #1.5. We further clearly stated that ensembles from this particular project have used for multiple other impact assessments* "The ISIMIP2b ensemble has already been used in multiple climate change studies investigating, e.g., flood risk (Willner et al., 2018; Thober et al., 2017; Alfieri et al., 2017), low flows in Europe (Marx et al., 2018), evapotranspiration (Wartenburger et al., 2018), runoff and snow in Europe (Donnelly et al., 2017) or multi-sectoral impacts (Byers et al., 2018)."

*We have updated this with recent publications and it now reads [105]:*

"The ISIMIP2b ensemble has already been used in multiple climate change studies investigating, e.g., flood risk (Willner et al., 2018; Thober et al., 2017; Alfieri et al., 2017), low flows in Europe (Marx et al., 2018), evapotranspiration (Wartenburger et al., 2018), runoff and snow in Europe (Donnelly et al., 2017), drought severity (Pokhrel et al., 2021), heat uptake by inland waters (Vanderkelen et al., 2020)  or multi-sectoral impacts (Byers et al., 2018; Lange et al., 2020)."

*Please see also our responses to #1.3 and #2.2.*

*We have clarified where information on the parameterisation can be found:*

[125 ff] "A comprehensive overview of GHMs and their properties can be found in Sood and Smakhtin (2014). Detailed model descriptions and evaluations of the models can be found in the primary publications referred to in the subsections below and Telteu et al. (2021) (for the model parameterisation see Sect. 2.2.)."

**1.8**

3. The outcome of the climate simulations should also be discussed in the paper to have an idea of how key forcing variables change over time and have a clear view of what should be expected in terms of groundwater changes.

*The primary variable driving the results is precipitation which was assessed for two RCPs in Fig 7. However, the GCMs from CMIP5 considered in the ISIMIP framework differ both in space and time and per variable. A proper assessment of the GCM output data is clearly out of the scope of this study and not necessarily connected with the groundwater recharge simulation in the hydrological model (as the GHMs differ in terms of input variable requirements). Such an assessment could only be determined with an extensive extra study. An extensive analysis of the sensitivity of groundwater recharge simulation to changes in climate input is of course very interesting but out of scope of this study but might be targeted in future research. See also #1.4.*

**1.9**

4. The authors discuss the trends they observed in the figures but there is no explanation about the drivers of these trends. One can expect to know why a region sees a high increase in groundwater recharge and other regions not. The sensitivity of a region to these changes should be discussed in this paper. These sensitivities are linked to the physical parameters of the region, these parameters aren't presented to it is really hard to understand the response of the region.

*Based on our general experience with the quantification of climate change impacts on hydrological variables, we believe that a region sees a high increase in groundwater recharge because there is a very high increase in precipitation, while without a rather high increase in precipitation, a region will not see any increase. However, to understand how groundwater recharge in different regions would react to the same changes in climate would require a different study setup. Such an analysis would require the concerted work of the various model developers and a well thought through sensitivity analysis setup, due to the complexity of daily simulations with multiple complex hydrological models (and different*

*climate scenarios). Truly knowledge-based statements about sensitivities go well beyond the scope of the paper (see also #1.4). This study is the first of its kind investigating the impacts of CO2 changes on recharge on a global scale, which is of interest to the research community (#1.1 the reviewer agrees) but has not been examined in other studies yet. It thus lays the groundwork for future studies that may include the research suggested by the reviewer.*

**1.10**

5. Differences observed between different models should also be discussed and explained, these differences may be related to the processes that these models are reproducing among others.

*We agree that the differences in model output relate to the differences in model implementation as discussed in the discussion section of the manuscript together with the comparison to Mohan et al. Dataset. Indeed, future research needs to investigate these differences further; this manuscript however is a start of such an assessment. We have contributed to a better understanding on how a changing climate impacts recharge by extensively discussing the effects of CO2 on the simulation of this process on large scales.*

**1.11**

6. Investigating the impact of CO2 on groundwater is a very important topic, and there is a lot of interest in the community to understand to what extent accounting for CO2 in hydrology impact the projected changes in groundwater. Nonetheless, as for the other results, the authors should provide an explanation about why a particular area sees a high impact and other areas not.

*We thank the referee for the agreement that investigating the CO2 effect is important. Nevertheless, we think that we have provided insights to the question raised by the referee. Figure 7 shows how particular regions differ in their response to modeled vegetation productivity. And the already existing paragraph below provides and explanation on why we see these differences:*

"Decreases in precipitation may lead to a decrease in vegetation productivity (if not counteracted by an increased water-use efficiency due to elevated CO2 concentrations (Singh et al., 2020)) and thus to a decrease in transpiration. GHMs assume shares for evapotranspiration (ET) in relation to potential ET and the available precipitation. In contrast, transpiration in CO2-driven models responds to active vegetation as well as the relations between different water flux components that simpler GHMs do not. This can explain why the dynamic vegetation models exhibit inter-model regional differences in the GWR response to P decrease. Further, some models (MATSIRO) may not calculate LAI (leave area index), which impacts transpiration. For models with active vegetation, the increase in water use efficiency due to stomatal conductance (also referred to as CO2 fertilisation) can compensate for the decrease in precipitation to some extent, making more water available for groundwater recharge as compared to the GHMs (Table 1).  Though in some regions, as seen in Figure 7 (and Fig. S10), this feedback is not enough to overcome the warmer and drier climate in terms of groundwater flux."

*Please also see our answer to #1.9 why a regional assessment of drivers is a challenging exercise that is not easily added.*

**1.12**

7. Given the types of models (not fully integrated hydrologic models) that the authors used, the uncertainties they are observing are likely tied to the uncertainties in the estimation of evapotranspiration. It would be great to discuss evapotranspiration first then the impacts on recharge and analyse how these two uncertainties differ.

*We agree that an assessment of uncertainties of AET in the GHMs and how changes in AET relate to changes in GWR would undoubtedly be interesting to understand the differences in model behaviors better. However, we do not believe that such a quantification of AET changes and their uncertainties would be helpful to those who need to adapt to climate-driven groundwater changes which is the proposed main audience of our study. What we do in our study to account for this important discussion is that we investigate the differences between models with active vegetation and without to make the reader understand that future changes in groundwater recharge strongly depend on estimates of AET.*

We added the line [463]: "Overall, the capability of a model to simulate actual ET largely influences its capability to simulate groundwater recharge."

**#2**

The manuscript deals with an important issue: the uncertainty on future groundwater recharge (GWR) due to climate change. This uncertainty is estimated using eights different global hydrological models (GHMs) and the outputs of four global circulation models (GCMs). I will not go into the debate of using GHMs instead of fully integrated physically based hydrologic models (IPHMs), even if the uncertainty would have been better estimated by using models with significant differences in their philosophy and conceptions. GHMs are valuable tools and the provided results are a good estimate of uncertainty in GWR at a global scale under this framework. Therefore, the manuscript is of a very good scientific level and suitable for publication in HESS.

*Thank you for the encouraging comment and the overall positive evaluation.*

**2.1**

1. The reliability of the GHMs is not convincingly described. A section should be dedicated to the description of the ability of GHMs to simulate real situations on different sites over the world. It should not be an exhaustive description of GHMs applications to real sites, but some examples found in the literature of different GHMs simulations of large scale watersheds under different climatic conditions. If GHMs are not able to estimate water balances including GWR properly at the time scale used in this work and over the last decades, the interest of this work appears to be limited.

*Model reliability regarding estimation on current global-scale groundwater recharge was evaluated by comparison to Mohan et al.. Other work has been investing the reliability of estimating groundwater recharge in GHMs as well e.g., for WaterGAP in Döll and Fiedler (2008) (also cited and discussed in this manuscript).*

*Reliability regarding future climate change driven changes in groundwater recharge cannot be investigated by comparison to observation data, and we assume in our study, as it is usually done in multi-model climate change assessments, that all models are considered to be equally reliable. Projected groundwater recharge changes certainly depend strongly on projected evapotranspiration changes, which is why we investigated the differences between GHMs that simulate the impact of active vegetation on evapotranspiration (thus responding to atmospheric $CO2$ changes) and those that do not.*

**2.2**

2. I missed GHMs parameters. Please provide an insight on models parameterisation.

*We thank the referee for pointing out that parameterisation is important. We have revised this by now stating more clearly where the model inputs are coming from and where more information can be found (see also our answer to referee 1).*

*Groundwater recharge depends on parameterisations of canopy, snow and soil water balances as well as, e.g., assumed equations for potential evapotranspiration, we cannot provide more details beyond what is provided in section 2.1 specifically for groundwater recharge. We did add a sentence as provided in our answer to comment 1.7.*

**2.3**

The last sentence in the abstract is awkward and has to be more specific. You cannot write '
[revised manuscript text omitted]

We thank the three reviewers for the positive feedback on the work we have invested in this manuscript and the detailed comments to further improve it. We certainly acknowledge the uncertainties involved in the presented results and recognize that the submitted abstract might create false expectations. We have thus rewritten the abstract as well as the conclusions and are now more precise about the explanation of statistically significant changes. Furthermore, we added a new table (Table 1) that provides an overview on which vegetation processes are implemented in which model, added a new Figure 2 that shows the ensemble mean, and extensively revised Figure 7 (former Fig. 5). We are certain that the presented manuscript is an important study of the capabilities and limitations of global hydrological models.

The following lists all comments of all three reviewers and our rebuttal in *italics*. Attached is a markup document that highlights the changes compared to the initially submitted document. Line numbers refer to the revised version of the document.

**Referee #1**

**1.1**

Although the title of the paper and the results emphasize model uncertainties, I think the primary result based on the analysis is that recharge is not very sensitive to climate change as only 15% of cells show significant increases or decreases in recharge based on pre-industrial baseline and only 8% of cells show
significant change in recharge from current 1 degree to projected 3 degree condition.

*It is not correct to conclude from the fact the only 15% of cells show significant changes that groundwater "recharge is not very sensitive to climate change". This misunderstanding by the reviewer is caused by the various meanings the term "significant" has. In our paper, with "significant" we do not mean that there are "large" changes. We mean a statistical agreement or rather non-agreement between the two ensembles of simulated recharge, the one consisting of recharge computed by the various models under e.g. pre-industrial conditions and the other consisting of recharge computed by the various models at a certain global warming level (as tested here by a Kolmogorov-Smirnov test). To avoid this confusion, we have clarified this fact in Section 2.5 and for multiple references to the significance of the changes. We have also added a new Figure 2 that shows the ensemble mean difference between a 3° C warming and the present day without any statistical tests. We have also strongly modified abstract and conclusions to clearly express that significant refers to statistically significant (see response 3.1).*

*Section 2.5 now reads (Line 258 ff):*
"A model ensemble allows us to consider the uncertainty in modeling physical processes as different model use different algorithms and parameters for computing groundwater recharge. To determine whether changes in GWR due to GW computed by the model ensemble are statistically significant, we use the two-sample Kolmogorov–Smirnov (K-S) test to compare the GWR values computed by all GHM-GCM model combinations under e.g., PI conditions with the values at the various GW levels. The use of a two-tailed t-test is not advisable in this setting due to the small sample size (max. 84 in this study). Because the K-S test does not allow to check whether the ensemble agrees on the sign of change in GWR, we apply an additional criterion to determine a significant change similar to Döll et al. (2018). A change is only marked as (statistically) "significant" if the K-S test indicates a significant difference and at

least 60% of the model realizations of the ensemble (RCP, GCM and GHM combinations) agree on the sign of change (i.e. a decrease or increase)". In case of a low significance, all models may show large responses to climate change while their agreement on the amount or sign of change is low. "

*The new Figure and its description (Line 270 ff):*

"To assess the impact of GW on GWR, Fig. 2 shows the ensemble mean change of GWR between the current 1°C world and a potential 3°C GW. We chose to express changes as absolute change rather than relative change because zero, or close to zero, GWR in some regions of the world leads to not defined or extremely large percentage increases and decreases (Fig. S1 and S2). The model mean shows large decreases of over 100 mm year-1 in South America and in the Mississippi Basin and decreases of up to 50 mm year-1 in the Mediterranean, East China, and West Africa. Increases of over 100 mm year-1 are prominent in Indonesia and East Afrika. Individual GHM-GCM model combinations compute much larger changes.

[Figure]

3 °C compared to present day (1 °C)

mm year$^{-1}$

−100 −50 −10 −1 0 1 10 50 100

**(new) Figure 2** Ensemble mean change in GWR [mm year-1] between conditions of present day warming of 1 °C GW and at 3 °C GW, averaged over the GWR changes of all GHM-GCM model combinations.

Ensemble mean changes as shown in Figure 2 may be low in some areas, but this could be due to large positive changes compute by some GHM-GCM model combinations being canceled by large negative changes by other model combinations. To assess the changes which show a high statistical agreement in-between the model combinations, we determine where computed changes of GWR are statistically significant (Section 2.5)."

**1.2**
It would be good to acknowledge that recharge is likely the most difficult component of the water budget to simulate because it is modeled as a residual, accumulating uncertainties in other water budget components.

*We agree and accordingly have changed the abstract and introduction. (Line 27 and 60):*
"Groundwater recharge is an important indicator for groundwater availability, but it is a water flux that is difficult to estimate as uncertainties in the water balance accumulate, leading to possibly large errors in particular in dry regions." And "The simulation of GWR is possibly one of the most challenging components of the water budget as it accumulates the uncertainties of all other components of the budget. Especially in semiarid regions, uncertainties in precipitation and evapotranspiration lead to considerable uncertainty in recharge. An additional factor in estimating groundwater recharge is the simulation of the groundwater table and thus capillary rise and focused recharge."

**1.3**

In addition, it is extremely difficult to simulate in semiarid regions because small uncertainties in precipitation and evapotranspiration can result in large uncertainties in recharge.

*Has been addressed together with 1.2.*

**1.4**

Many studies suggest that climate change will result in increased climate extremes (floods and droughts) that may result in increased recharge from focused rather than diffuse recharge; however, it seems that few of the models consider focused recharge.

*We agree, however none of the models includes a reliable implementation of focused recharge. Current developments of global gradient-based groundwater models will improve the implementation of these processes further but currently we focus on diffuse GWR in this study.*

*This is a limitation now stated more clearly (Line 125):*
"We do not consider focused recharge in this study as no model offers a reliable implementation of these processes until now."

**1.5**

The authors refer to groundwater levels throughout the paper with respect to temperature levels; however, this is confusing as groundwater levels are generally considered water table levels. It might be good to include temperature when referring to these.

*The submitted manuscript makes no assumptions about the effect of changes in groundwater recharge on groundwater levels. We assume that the referee misread parts of the manuscript as the abbreviation GW (Global Warming) can easily be confused with GroundWater in GWR (Groundwater Recharge). While we realize this is not a perfect choice for an abbreviation we chose to keep it.*

**1.6**

I agree that it is good to focus on absolute changes in recharge rather than relative changes. The authors suggest that underestimating runoff would result in increased GWR; however, this would not
be the case if GWR is focused and derived from runoff as in semiarid regions (L. 74).

*We are happy to hear that the reviewer agrees with our choice in using absolute changes. This is something we have debated extensively when writing this manuscript. We agree with the comment, however as stated in 1.4 this study mainly focuses on defuse groundwater recharge.*

**1.7**

The authors repeatedly use present tense to refer to work that was done for this study. I think it would be more appropriate to use past tense.

*The manuscript was heavily revised in this regard at multiple places and should now contain a more precise use of tenses. Please see the attached markup document.*

**1.8**

The model CLM-5 has been upgraded substantially relative to CLM-4.5. It might be good to consider CLM-5 rather than CLM-4.5.

*It is true that CLM-5 is an improved version of the model, however our analysis is based on the available ISIMIP 2.b outputs, where only the modeling team of CLM-4.5 has submitted groundwater recharge.  Running additional simulations with another model would compromise the reproducibility of our findings and clear link to rigorous ISIMIP protocol. This study is not an investigation of a specific model but of a consistent model ensemble.*

**1.9**

Soil thickness varies substantially among the models (e.g. LPJ 13 m thick). It would be good to comment on the impact of varying soil thickness on model results.

*While we agree that such an analysis is of interest it is clearly out of scope for this paper. It would require to modify and rerun all the complex models in a sensitivity analysis. The corresponding author is not the developer of these models, which are very complex to setup and run and even harder to compare, this is why we are using the ISIMIP framework, which enables a baseline that allows for the complex model comparison shown in this manuscript. To allow for a more in-depth comparison on how different processes are implemented in GHMs there is currently a complex manuscript under development targeting the implementation differences in these models.*

**Referee #2**

**2.1**

One of the main conclusions is that dynamic vegetation has a strong impact on the estimated GWR, which is far from being as obvious based on the results presented.

*While we agree that there is a significant uncertainty on how changes in CO2 levels impact the water balance Figure 7 and 8 clearly shows substantial differences between the model types. Of course, this does not necessarily mean that this difference can only be explained through the simulation of dynamic vegetation. Nevertheless, it seems likely and supports that further research on this topic is necessary.*
*We added a sentence to make this clearer in our conclusions (Line 579 ff):*

"However, consideration of these processes does not lead to a uniform increase of groundwater recharge, as might be expected from the physiological effect of increasing atmospheric $CO_2$ concentration. In some regions with decreasing groundwater recharge, where groundwater availability is a major concern, models that include these processes show the largest differences among themselves. Further research is necessary to understand GWR on large scales, and how it is affected by climate. Simulation of groundwater recharge by global hydrological models needs to be analyzed in more detail, and the benefit of integrating gradient-based groundwater flow models in GHMs should be assessed."

**2.2**
In addition, some figures are difficult to read and seem to support only partially the comments.

*We have improved the font size in multiple figures. See also comment 2.6.*

**2.3**
Introduction: The choice is made to estimate ground water recharge throughout the continental part of the world. However, many areas have no extended groundwater. Whymap.org provides map of the extension of the aquifer. What does it mean to estimate GWR where there is no or local and shallows aquifer? Won't it be more interested to estimate the recharge on the aquifer domains?

*Groundwater recharge is also of interest outside of the major global aquifers represented in WHYMAP in blue. In each 0.5° grid cell there are very likely local aquifers (e.g. in alluvial valleys, and confined aquifers are also affected by groundwater recharge. There are other studies which use WHYMAP e.g. Gleeson et al. 2012 Nature, or Taylor et al. 2013 NCC, however our analysis here firstly focuses on a grid-based analysis (allowing for a better understanding on the model differences), which could be in a follow-up study be applied on an aquifer scale.*

**2.4**
Line 200: It is necessary to provide information on the bias-adjusted method? What is the assumption? What is the reference climate used?

*A detailed explanation of the method is out of scope of the paper. We have added, to the existing reference to additional information on the climate inputs, a reference of the used method (line 213):*

"The bias adjustment method used for the GCMs in ISIMIP2b is using a trend preserving algorithm (Frieler et al., 2017) with EWEMBI (Lange 2018) as baseline (reference) climate condition."

**2.5**

Section 2.1: I suggest to provide a table that summarizes if GHM includes or not in the GWR another part that a partition of the precipitation, especially, which GHM includes river to aquifer exchange, since this may be a very important difference in alluvial regions. Moreover, this table should also summarize which GHM integrates direct effect of CO2, explaining clearly if they account for stomatal aperture sensitivity to CO2 and/or for vegetation dynamic (LAI) (so far, this is not clear).

*As explained in 1.4 we do not consider focused recharge or transmission losses in this study (see 1.4). Also, it might be difficult to distinguish specific alluvial regions on this coarse resolution. A new Table 1 has been added to summarize the implementation of CO2 related vegetation processes more clearly (Line 126 ff).*

**Table 1** Overview which models are able to simulate the impact of evolving $CO_2$ concentrations on vegetation and how it is implemented.

| GHM | Considers $CO_2$ | Summary of considered vegetation processes in ISIMIP2b | Reference |
|---|---|---|---|
| WaterGAP2 | No | - | - |
| CLM4.5 | Yes | Photosynthesis depends on root zone soil moisture availability. The description is similar to LPJmL listed below. The area a population of plant functional types (PFTs) takes up is prescribed and only changes if the input data does. | (Di Liu and Mishra, 2017) |
| H08 | No | - | - |
| JULES-W1 | Yes | Evapotransipration is considered from five PFTs and four non-vegetative surface types. Each grid cell is composed of different fractions of those 9 surface types. Transpiration occurring from vegetation is based on photosynthetic process, which is subject to stomatal conductance regulated by the $CO_2$ concentration. Furthermore, transpiration is also controlled by the soil moisture availability in the root zone. | (Best et al., 2011; Clark et al., 2011) |
| LPJmL | Yes | Vegetation composition is determined by the fractional coverage of PFTs at the grid scale. PFTs are defined to account for the variety of structure and function within a stand and are therefore simulated as average individuals competing for light and water according to their crown area, LAI, and rooting profiles. The vegetation dynamics component of LPJmL includes carbon allocation to different PFT tissue compartments, PFT interaction, and establishment and mortality processes. Photosynthesis and stomatal response are simulated following Farquhar et al. (1980) and the generalization by Collatz et al. (1991) for global modelling, based on the function of absorbed photosynthetically active radiation, temperature, day-length, and canopy conductance for each PFT present in a grid cell. | (Schaphoff et al., 2018) |
| PCR-GLOBWB | No | - | - |
| CWatM | No | - | - |
| MATSIRO | Yes | The consideration of $CO_2$ effects is functionally similar to that in CLM, and there is no dynamic vegetation scheme. $CO_2$ is prescribed in the model, which is used in the | (Takata et al., 2003) |

photosynthesis scheme to calculate stomatal conductance, among other parameters, following Farquhar et al. (1980). Soil moisture stress on photosynthesis is considered using moisture availability in the root zone with root distribution fraction in each soil layer. All of that is done for different vegetation or plant functional types.

**2.6**

Figure 2 is of bad quality.Is it necessary to have all the extremes? Is it reasonable to have a range over 2000mm/year? most aquifers recharge maps stop before 1000mm/year and often under 500. Are GWR values above 1500 in Figure 2 located in capacitive aquifers or in very local and shallow aquifer as defined by Whymap (see comment 1)? As this figure is difficult to read, it is impossible to check comments line 317-322

*We think it is necessary to include all extremes even if it reduces the readability of the figure. Only by showing the outliers we are able to openly discuss what the models compute no matter if the values are reasonable or not. Again, the focus on specific aquifers is not something we target in this manuscript (See also 2.3).*

**2.7**

Nice to disentangle the impact of GCM and GHM, but, fig3a includes 76 cases while fig b includes only 36 cases. How does this compare? In order to try to understand what the impact of GHM is, and what the impact of the response of GHM to GW is, it seems required to show the variance of the 8 GHMs on preindustrial case. Figure 7 shows that there are important change, but, variance will be helpful to compare with fig3a.

*As explained in 2.3 not all RCPs and GCM combinations may lead to a warming of 3°, thus the number of involved ensemble members changes. The simulation of PI GWR per GHM is already shown in S1 and S4. We now also refer to these figures in this paragraph: (Line 393)* "For the simulated variance at PI see Fig. S1 and S4."

**2.8**

Figure 4 includes only one realization of one GHM and one GCM. Why this GCM? Why this GHM? Is this GCM includes dynamic vegetation? Is the dynamic vegetation of the GCM consistent with the one of the GHM? In any case, it would be nice to have some information on LAI changes;

*This is stated at the end of the paragraph* "Unfortunately, no other GHM-GCM combinations with these alternative $CO_2$ concentration variants are available in the framework of ISIMIP2b." *Yes, GFDL includes dynamic vegetation as well. We agree that an assessment of the differences between the GHM and GCM implementation of the dynamic vegetation would be of interest, however, it is clearly out of scope of in this study. It is also unclear to us how information on LAI changes would provide more insights on the presented results.*

**2.9**

Line 422 : Decrease of precipitation lead to a decrease of vegetation productivity⇒I guess it is more complicated than that, you may correct

*Agreed it very much depends. We rephrased the sentence and added another relevant reference in this matter.*

*Now reads (Line 458 ff):*
"Decreases in precipitation may lead to a decrease in vegetation productivity (if not counteracted by an increased water-use efficiency due to elevated $CO_2$ concentrations (Singh et al.; 2020))) and thus to a decrease in transpiration."

**2.10**

Figure 5: is this figure correct? a same GHM can appear twice for a subregion with the same colors▷A change of precipitation of 100mm/year can lead to an increase of GWR of the same amount▷ Is the vegetation dead? It would be nice to better understand which process occurs in detail in one subregion?

*Yes, the figure is correct. However, we realize that it might be difficult to distinguish between the amounts of markers. To allow for a more comprehensive figure we revised in heavily and summarized the models in a bar chart (see new Figure 7 below). We have furthermore improved the original plot and added it to the supplement to allow the interested reader to still investigate the model differences. Concerning the large increases of GWR linked to large increases of precipitation, it is unclear which process is mainly responsible for this feedback. Possibly the increased water use efficiency of the vegetation allows for a large increase in GWR linear to the increase in P.*

[Figure]

[Figure]

**(heavily revised) Figure 1** Relation of changes in precipitation (P) (mean(1981-2010) – mean(2070-2099)) to changes in GWR (mean(1981-2010) – mean(2070-2099)) depending on the model type (with or without $CO_2$; see also Table 1) per SREX (selection as in Table 3)  for RCP 2.6 and RCP 8.5 for the GCM HadGEM2-ES.

[Figure]

**(new) Figure S10** Relation of changes in precipitation (P) (mean(1981-2010) – mean(2070-2099)) to changes in GWR (mean(1981-2010) – mean(2070-2099)) depending on the model type per SREX (selection as in Table 1) for RCP 2.6 and RCP 8.5 for the GCM HadGEM2-ES. Y-Achis is log-scaled. The dashed line is the 1:1 line.

**2.11**

Fig 6 compares 4 GHM with dynamic vegetation (change in LAI?) with the 4 others who assumes constant vegetation (but do they account sensitivity of the stomatal aperture to CO2?). The difference on the physics of the GHMs is large, but the impact seems reduced. How can we be sure that the difference is linked to the dynamic vegetation? Similar difference may well exist between this different GHMs without change in vegetation dynamic▷ Moreover, are the significant changes located where the aquifers are extended and capacitive, or where the aquifers are very shallow and local?

*No, the other 4 models with constant vegetation do not account for the stomatal aperture. To clarify we now define the terminology much more clearly in 2.1 to avoid any confusion. Additionally, we added Table 1 to provide an overview on which model implements what kind of vegetation.*

Now reads in S2.1:
"In the following, we use the term *active vegetation* for models that consider the physiological effect of changes in $CO_2$ on vegetation and the term *dynamic vegetation* for the models that allow for a changing vegetation regarding LAI and/or vegetation type."

*We agree that we cannot be sure without a further extended sensitivity analysis, which unfortunately is not possible at this point, that the differences we are seeing are solely due to the inclusion of vegetation. However, the only analysis possible in this regard with the available data shown in Figure 6 (based on new version) supports that assumption. We are now stating this more clearly:*

Line 493: "It is likely that the shown differences are due to the implementation of dynamic vegetation in the GHMs (compare Fig. 6), however it is possible that other model peculiarities and processes are relevant as well."

*For your comment on specific aquifers see 2.3.*

**2.12**

Assessment of the GHM should not appear in the discussion. It should be earlier, or in supplement.GWR estimates by Mohan is only the GWR from precipitation (no river inputs). How does this compare to the numerous GHM?

*We don't think that an earlier appearance is merited. The assessment of the models themselves is not the focus of this paper, thus a discussion of the figure at the beginning of the paper is not useful to convey our central messages. Moving them to the supplement is also not helpful as Fig. 9 transports important messages to understand the relevance of the presented results and conclusions correctly. We think that it is a helpful figure to discuss the limitations of the approach without skewing the central message of the paper. The discussion is thus the right place even though it might be unusual.*
*Concerning the question if the estimates are comparable see also our answer in 1.4.*

**2.13**

Line 539: "Despite the uncertainties, this study shows that climate change will impact groundwater availability in many regions of the world". this is naïve: it was already shown by numerous regional studies.

*Additional evidence is always a good thing. We further provide three important additional messages that regional studies are not able to show: 1) We provide global patterns of change – also for regions that have not been studied yet and consistent patterns on much larger scales. 2) We present a multi-model ensemble approach, which is also not common in regional studies. 3) The used models operate at the coarse spatial resolution closer to climate models without the requirement to downscale uncertain climate input to specific regions.*

We rephrased the conclusion and it now reads:
"Potential GWR changes due to climate change require increased attention from the scientific community as well as from decision-makers because they affect future water availability in many regions and thus the wellbeing of billions of people. This study shows that simulated global-scale estimates of GWR vary strongly among GHMs, which contribute more strongly to the overall uncertainty of future groundwater recharge than the applied GCM output. However, statistically significant increases and decreases of GWR could be identified in specific regions per GW level. The presented inter-model ranges of GWR changes are an important input for processes aiming at developing strategies for climate change adaptation, as risk-averse decision-makers may want to orient their strategies towards adapting to the worst-case GWR change and not to the projected ensemble mean change.

This study shows that including vegetation processes in GHMs can change projected GWR changes substantially. However, consideration of these processes does not lead to a uniform increase of groundwater recharge, as might be expected from the physiological effect of increasing atmospheric $CO_2$

concentration. In some regions with decreasing groundwater recharge, where groundwater availability is a major concern, models that include these processes show the largest differences among themselves. Further research is necessary to understand GWR on large scales, and how it is affected by climate. Simulation of groundwater recharge by global hydrological models needs to be analyzed in more detail, and the benefit of integrating gradient-based groundwater flow models in GHMs should be assessed."

**2.14**

Line 545: "Moreover, this study shows that including dynamic vegetation processes in GHMs can change the results substantially": this is not that clear from your results. I was expected more impact indeed▷

*This relates to 2.5, 2.8, 2.9, 2.11 and should be much clearer now.*

**2.15**

A map of the extension of the subdomain is required

*We assume that the referee is referring to the SREX regions, which are provided in Fig S6.*

**2.16**

Numbering of figures is wrong

*Figure 1 existed twice in the manuscript. We apologize for any caused inconvenience. This was corrected.*

**2.17**

Line 287: Masson-Delmotte et al

*This has been corrected.*

**2.18**

Figure S3 should be in the text since it is discussed

*We disagree. It is only shortly mentioned and not necessary to understand to message of the paper. It would be possible to move it to the Appendix but we don't think that it adds any valuable information that is not already transported through the text.*

**Referee #3**

**3.1**

I do appreciate the tremendous amount of work the authors put into the study and already apologize for not being able to be more positive. The study has a flaw in that it assumes/postulates that GHMs are able to simulate groundwater recharge processes (line 61). This remains to be shown (see specific comment below). The authors also compare to Mohan et al. (2018), a data set, which is also highly

uncertainty itself, and show that all models essentially have no skill. Thus, the study is hypothetical and should be seen as a model sensitivity study, which does not necessarily reflect reality. The large uncertainty in the results supports that notion. The authors discuss the limitations in detail and come to the conclusion (line 538) "Despite the uncertainties, this study shows that climate change will impact groundwater availability in many regions of the world. " Yes, that's probably true. But it's sad that due to the large uncertainty, no additional concrete conclusions can be drawn from the results.

Because the detailed analyses and numbers have very low confidence, I am not able to comment on the simulation results. In summary, I find that GHMs have not been tested comprehensively especially with regard to complex processes such as groundwater recharge. Thus, in my opinion, the study is too early (first testing, then analyses). Looking at the description of the models, I would not even call the estimated flux groundwater recharge. If GHMs are applied I suggest to study the major fluxes of the water balance including their impact on the residuals of the balance equation in the models. In a way that's what the authors are doing, yet the presentation does not show these results.

*We appreciate the recognition of the amount of work that went into this study and strongly disagree with the conclusion of reviewer #3 that "the study is too early". We think it is timely to do a study that aims at understanding the best information we have on potential impacts of climate change on groundwater at the global scale, which is provided by the multi-model ensemble output that is analyzed in this study. The merit of our study is similar to that of many climate change studies done with global climate models. It is well known that different global climate models project very different future climatic changes in response to the same greenhouse gas emissions scenario; nevertheless, their results including an understanding of their uncertainties are of interest to many and impact decision-making. We explicitly wanted to focus on uncertainties and not (impossible) predictions, which is also reflected by the title of our manuscripts: ""Uncertainty of simulated groundwater recharge ...".*

*It is definitely justified to criticize the ability of GHMs to simulate groundwater recharge, which we openly discuss in this publication. The referee himself/herself admits that "The authors discuss the limitations in detail". While the analysis has flaws the study discusses them openly and it provides new information and understanding. That no precise and certain prediction of future groundwater recharge are currently possible is certainly, using the word of the referee "sad". However, the study does identify statistically significant increases and decreases in recharge in multiple regions of the world. And it provides a range of potential future recharge changes for each world region that persons in charge of climate change adaptation should take into account for lack of better knowledge.*

*The statement that "that GHMs have not been tested comprehensively" contradicts a substantial number of studies that have been published in the recent years devoted to the evaluation of GHMs, for example: (Scanlon et al., 2018; Müller Schmied et al., 2014; Döll and Fiedler, 2008; Döll and Flörke, 2005) and many more. Still, additional testing and improvement is still necessary and this study provides important pointers for future research. In any case, we agree that groundwater recharge is a complex process that needs to be developed further in GHMs. Our manuscript offers first insights on what these improvements might be by presenting a novel comparison of a large ensemble of models in this regard. The suggested approach of comparing other components of the water balance is something that has been done in the framework of ISIMIP in multiple other publications ("e.g., flood risk (Willner et al., 2018; Thober et al., 2017; Alfieri et al., 2017), low flows in Europe (Marx et al., 2018), evapotranspiration*

(Wartenburger et al., 2018), runoff and snow in Europe (Donnelly et al., 2017) or multi-sectoral impacts (Byers et al., 2018).") *that are cited in the introduction of this manuscript.*

*In reaction to the reviewer comment, we made substantial changes to both abstract and conclusions to focus less on specific numerical values than on explaining the potential benefits and applicability of the study results.*

*The new abstract:*

[revised manuscript text omitted]

**3.2**

Only a couple specific comments Abstract: The reported percent increases/decreases of GWR suggest an accuracy that is simply not there; especially given the huge uncertainty in the results. Thus, the abstract sends the wrong message, especially to water managers and decision makers.

*We agree that there should be no confusion on the main message of this manuscript and that the results should be considered with care. To attract the reader's attention on this matter and avoid any confusion for readers that might misinterpret the results we have greatly adapted the abstract. See also 3.1..*

**3.3**

54: One of the most important factors is missing: Depth and dynamics of the free water table.

*This is now mentioned (Line 63 ff):*

"An additional factor in estimating groundwater recharge is the simulation of the groundwater table and thus capillary rise and focused recharge. This has not been achieved yet in GHMs, however, recently, global hydrological models (GHMs) started integrating gradient-based groundwater models to better estimate the flows between surface water and groundwater as well as the impact of humans and the changing climate on the groundwater system (de Graaf et al., 2019; Reinecke et al., 2019). Neglecting capillary rise may lead to an overestimation of decreases and increases of GWR due to a changing climate."

**3.4**

61: This is true, but has never been tested. I suggest to compare the GHMs against fully integrated hydrologic models, such as Cathy, Hydrogeosphere, OpenGeosys, ParFlow, etc. in order to test the ability of GHMs to simulate recharge processes. This is one of many tests that GHMs should undergo in my opinion.

*The models currently under development have been compared to these models (e.g. ParFlow in Reinecke et al. 2019). We agree that this comparison needs to continue in the future.*

*We added a sentence to the conclusions to reflect that. See 3.1.*

**3.5**

Section 2.2: Porbably I missed these details: what's the time step, the spatial resolution, etc?

*The spatial resolution is 0.5° x 0.5° (described in section 2.1) and temporal resolution of the original GWR data is monthly, which was averaged to yearly values in this study (also described in 2.1). Time step is a term one might use in the context of numerical models, such as gradient-based groundwater models, which are not part of this study.*

**References**

[revised manuscript text omitted]